# Tumor-targeted glutathione oxidation catalysis with ruthenium nanoreactors against hypoxic osteosarcoma

Hanchen Zhang[1,2], Nicolás Montesdeoca[3], Dongsheng Tang[1,2], Ganghao Liang[1,2], Minhui Cui[1,2], Chun Xu [4], Lisa-Marie Servos[3], Tiejun Bing[5], Zisis Papadopoulos[3], Meifang Shen[6], Haihua Xiao [1,2] ✉, Yingjie Yu [6] ✉ & Johannes Karges [3] ✉

The majority of anticancer agents have a reduced or even complete loss of a therapeutic effect within hypoxic tumors. To overcome this limitation, research efforts have been devoted to the development of therapeutic agents with biological mechanisms of action that are independent of the oxygen concentration. Here we show the design, synthesis, and biological evaluation of the incorporation of a ruthenium (Ru) catalyst into polymeric nanoreactors for hypoxic anticancer therapy. The nanoreactors can catalyze the oxidation of glutathione (GSH) to glutathione disulfide (GSSG) in hypoxic cancer cells. This initiates the buildup of reactive oxygen species (ROS) and lipid peroxides, leading to the demise of cancer cells. It also stimulates the overexpression of the transient receptor potential melastatin 2 (TRPM2) ion channels, triggering macrophage activation, leading to a systemic immune response. Upon intravenous injection, the nanoreactors can systemically activate the immune system, and nearly fully eradicate an aggressive osteosarcoma tumor inside a mouse model.

Osteosarcoma is the most common aggressive malignant tumor originating from bone tissue in children and adolescents. It is characterized by rapid progression, a poor prognosis, and early metastasis. This solid tumor arises from bone tissue and destroys healthy bone structures. Solid tumors, such as osteosarcoma, generally suffer from an inadequate oxygen supply due to the rapid proliferation of cancer cells, incomplete vascularization, and uneven distribution. This results in oxygen-deprived areas within the tumor tissue. In the core, a necrotic nucleus forms due to the prolonged lack of oxygen and nutrients. The necrotic nucleus extends outward into a region with low oxygen concentrations, referred to as hypoxia. The hypoxic tumor microenvironment is considered one of the biggest obstacles for the development of anticancer drugs. The vast majority of compounds show a reduced or even complete loss of the therapeutic effect under hypoxic conditions. As the primary factors for the reduced anticancer effect 1) the reduced delivery and uptake of the anticancer agent, 2) the need for the conversion of molecular oxygen ($O_2$) into reactive oxygen species (ROS), 3) the preferred therapeutic effect against highly proliferating cells, which is reduced under hypoxic conditions, or 4) adaptation of the cancer cells by promoting cell survival and upregulating certain resistance pathways, has been suggested[1-5]. To overcome these limitations, there is a need for the development of anticancer agents that can therapeutically intervene in hypoxic cancer cells.

Among the most promising therapeutic approaches, much research interest has been devoted to catalytic metal-based anticancer agents that could convert natural biomolecules into (cytotoxic)

[1]Beijing National Laboratory for Molecular Sciences, Laboratory of Polymer Physics and Chemistry, Institute of Chemistry, Chinese Academy of Sciences, Beijing, China. [2]University of Chinese Academy of Sciences, Beijing, China. [3]Faculty of Chemistry and Biochemistry, Ruhr-University Bochum, Bochum, Germany. [4]School of Dentistry, The University of Queensland, Brisbane, QLD, Australia. [5]Immunology and Oncology center, ICE Bioscience, Beijing, China. [6]State Key Laboratory of Organic-Inorganic Composites, Beijing Laboratory of Biomedical Materials, Beijing University of Chemical Technology, Beijing, China. ✉e-mail: hhxiao@iccas.ac.cn; yuyingjie@mail.buct.edu.cn; johannes.karges@ruhr-uni-bochum.de

products or products that ultimately lead to cell death. These types of catalytic transformations could be broadly classified into oxidation and reduction, hydrogen transfer, bond cleavage, or cyclization reactions[6–11]. Redox reactions typically utilize the biologically accessible oxidization states of the metal core to transfer electrons to naturally occurring substrates that are essential for the cancer cell metabolism and therefore induce cell death. Previous studies have demonstrated the oxidation of gluthathione (GSH) into glutathione disulfide (GSSG)[12], the oxidation of NADH into NAD$^+$[13], or the reduction of azides into amines[14]. Transfer hydrogenation reactions reduce naturally occurring substrates inside the cancer cells in the presence of a catalyst and a natural hydride source. Previous reports have shown the reduction of aldehydes into alcohols using NADH as a hydride source[15], NAD$^+$ into NADH using formate as a hydride source[16], pyruvate into lactate using formate as a hydride source[17,18], or ally alcohols into ketones[19]. Protecting groups as well as terminal functionalization could be removed through transition metal complex catalysis. Previous studies have demonstrated the cleavage of allylcarbamates[20], propargyls[21,22], or allenes[23]. The transition metal complex could also catalyze the cyclization inside cancer cells. Previous reports have indicated intramolecular hydroarylation[24], ring closure metathesis[25], cyclo addition[26,27], or cross coupling[28] reactions. Despite recent developments in the field of metal-based catalytic drugs, these compounds are generally associated with a poor spatial and temporal control of the treatment as well as poor cancer selectivity, resulting in potential side effects and non-ideal drug doses[29].

To overcome these limitations, in this study, we describe the rational design, synthesis, and in-depth biological assessment of ruthenium (Ru)-containing nanoreactors for the catalytic oxidation of GSH to GSSG in hypoxic osteosarcoma. Using density functional theory calculations, the catalytic cycle of the ruthenium catalyst is predicted. Experimental investigations demonstrate the capability of the catalyst to convert GSH in GSSG. To enhance the therapeutic potential of the ruthenium catalyst and tackle challenging hypoxic tumors, the metal complexes is covalently linked to the side chain of a polymer that contains a carbon fluorinated moiety capable of delivering oxygen to the tumor site. Based on the amphiphilic nature, the polymer can self-assemble into tumor-targeting nanoreactors. The ruthenium catalyst and carbon fluorinated moiety-loaded nanoreactors are found to be therapeutically active against hypoxic osteosarcoma cells. An analysis of the mechanism reveals that the nanoparticles facilitate the oxidation of GSH into GSSG, leading to disruptions in redox balance and the antioxidant defense system and therefore accumulation of ROS and lipid peroxides. Simultaneously, catalytically produced hydrogen peroxide (H$_2$O$_2$) activates the protein transient receptor potential melastatin 2 (TRPM2) ion channel, promotes calcium influx, activates macrophages, and induces systemic immune response (Fig. 1). Upon intravenous injection, the nanoreactors selectively accumulate in the tumor and nearly fully eradicated an aggressive osteosarcoma inside a mouse model.

## Results
### Design, molecular synthesis, and characterization
Various studies have described the ability of transition metal complexes to act as catalysts for organic and inorganic transformations. Recently, increasing attention has been focused on the application of metal complexes to catalyze the transformation of biological essential biomolecules into (cytotoxic) products or products that ultimately lead to cell death. Typically, within the activation step, the metal-halide bond is cleaved, and the open coordination site of the metal complex is used for electron or proton transfer reactions. Despite promising preliminary findings, these compounds are generally associated with a poor spatial and temporal control of the treatment as well as poor cancer selectivity, resulting in potential side effects and non-ideal drug doses.

To overcome these limitations, herein, the tumor-selective transport of a physiological stable ruthenium-based catalyst for the oxidation of GSH into GSSG as a nanoreactor is proposed. The design of these nanoreactors is based on: 1) Stabile ruthenium catalyst. While chloride ligands are typically quickly exchanged under physiological conditions, bromo and in particular iodo ligands are relatively inert to hydrolysis[30,31]. The bidentate phenylazopyridine ligand enables efficient electron transfer reactions while remaining tightly bound to the metal center. Previous studies of metal complexes with a phenylazopyridine ligand showed that this ligand could reversibly accept as well as donate two electrons in one electrochemically accessible, lowest unoccupied molecular orbital[32]. Capitalizing on this, a ruthenium complex with an iodo, p-cymene, and phenylazopyridine ligand Ru(II)-OH was designed. 2) Functionalization of the catalyst. To improve the pharmacological properties and enable tumor-selectivity, the phenylazopyridine ligand was terminally functionalized with a hydroxy group that could be used for further chemical modifications. 3) Reactive oxygen species (ROS) sensitive linker. The linker 2,2′-(propane-2,2-diylbis(sulfanediyl)) bis(ethan-1-ol) is able to rapidly decompose in the presence of ROS, enabling the biocompatibility and biosafety of the biomaterial. 4) Transportation of molecular O$_2$. The carbon fluorinated polymeric unit, that is able to physically entrap molecular O$_2$, was incorporated into the polymer backbone for selective delivery of O$_2$ to the target tissue. This could enable the treatment of hypoxic tumors, which represent a major challenge in anticancer drug development. 5) Enhancement of the blood circulation time. Carboxyl groups were incorporated in the polymeric backbone to promote long blood circulation times. 6) Amphiphilic nature of the polymer. The polymeric material is able to self-assemble into nanoreactors. 7) Terminal functionalization with polyethylene glycol. The polyethylene glycol chains improve the physiological stability and water solubility. 8) Selective tumor accumulation. Based on the enhanced permeability and retention effect, the nanoparticles can accumulate in the tumorous tissue[33,34]. The incorporation of a ruthenium catalyst and carbon fluorinated moieties into polymeric nanoreactors is expected to provide an anticancer strategy for tumor-targeted catalytic anticancer therapy of hypoxic tumors.

Using density functional theory calculations, the catalytic cycle of the transformation of GSH to GSSG using Ru(II)-OH as a catalyst was calculated (Supplementary Tables 3–6). During the initial step, the thiol group of GSH is conjugated to the azo double bond of Ru(II)-OH, causing a reduction to a hydrazo moiety. Previous reports have demonstrated the occurrence of the conjugation of GSH to azo esters[35] or diamides[36]. Upon binding of a second equivalent of GSH into a highly energetic intermediate, the oxidized product GSSG is formed, and a ruthenium diamine complex generated. These products were found with a reduced energy than the corresponding starting materials, indicative of an energetically favored catalytic transformation. In the presence of O$_2$, the catalyst Ru(II)-OH is regenerated. (Proposed catalytic cycle: Fig. 2a, energy profile: Fig. 2b). The reoxidation of compounds with iodine containing molecules in the presence of O$_2$ through hydrogen peroxide (H$_2$O$_2$) intermediates has been previously demonstrated[37]. Notably, the azo-containing chemotherapeutic drug procarbazine for the treatment of Hodgkin's lymphoma and brain cancers was found to interact through a similar mechanism of action involving the transformation of GSH to GSSG and regeneration of the catalyst through a reaction with O$_2$[38,39], providing feasibility of the proposed catalysis. Combined these findings suggest the potential application of Ru(II)-OH as a catalyst for the transformation of GSH to GSSG.

Based on this theoretical insight, the catalyst Ru(II)-OH was prepared. The phenylazopyridine ligand was synthesized from hydroquinone and 2-hydrazine pyridine in the presence of perchloric acid. Upon nucleophilic attack of the phenolic oxygen atom at bromopropanol, the ligand was extended with an aliphatic terminal hydroxy

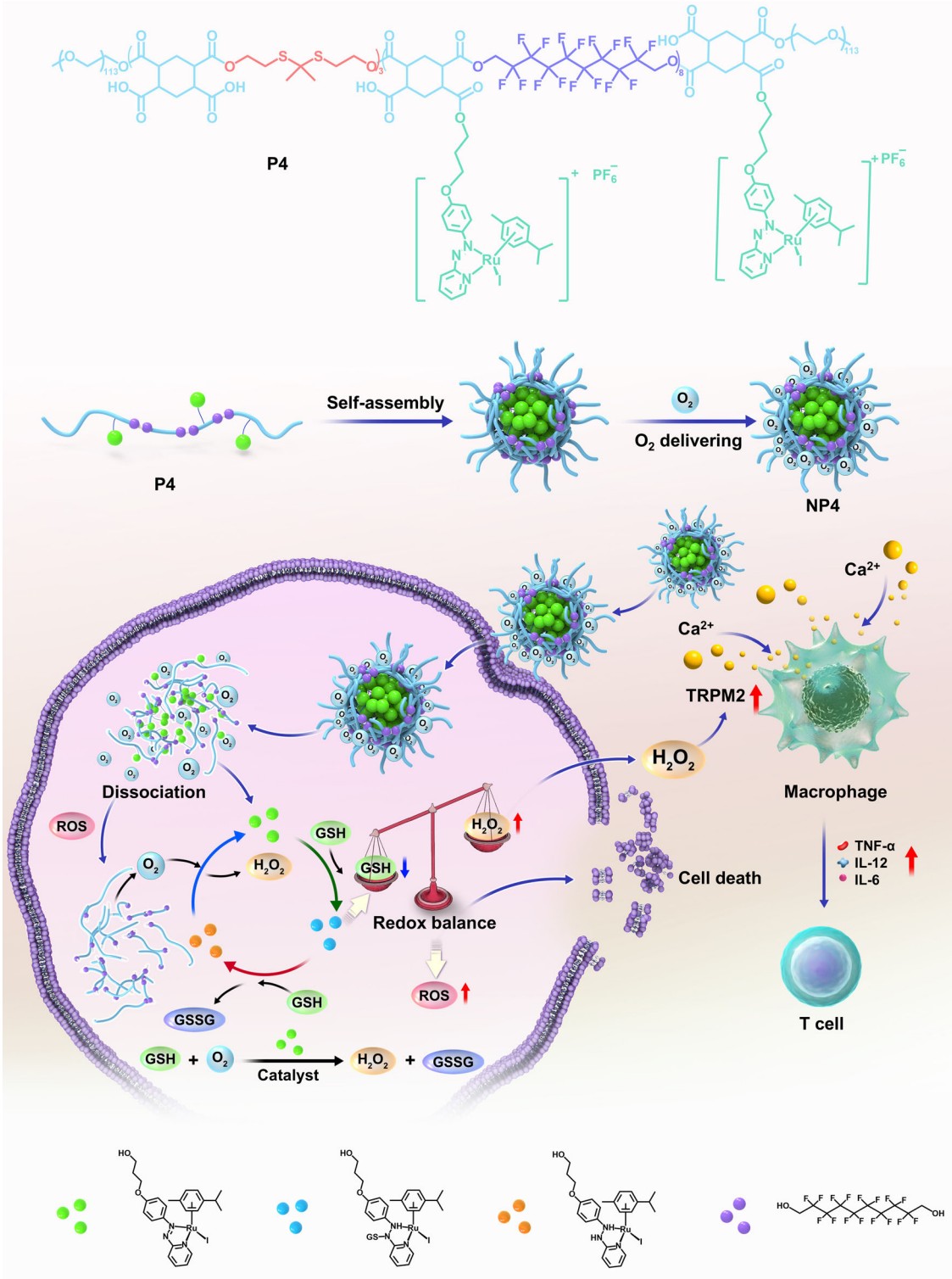

**Fig. 1 | Structure and biological mechanism of action of NP4 for tumor-targeted glutathione (GSH) oxidation catalysis of hypoxic osteosarcoma tumors.**

group. The ruthenium arene precursor was prepared from a mixture of p-cymene and ruthenium trichloride. Upon heating of this compound in an excess of potassium iodide, the iodo ruthenium arene precursor was formed. The reaction of the phenylazopyridine ligand and the iodo ruthenium arene precursor yielded the desired ruthenium catalyst Ru(II)-OH (synthetic strategy: Supplementary Fig. 1). The compounds were characterized by nuclear magnetic resonance (NMR) spectroscopy and high-resolution mass spectrometry (Supplementary Figs. 5–10).

## Catalytic activity

The ability of Ru(II)-OH to interact or convert GSH was studied upon time-dependent monitoring of an aqueous solution by $^1$H-NMR spectroscopy. After incubation for 12 h, a new peak at 3.26 ppm appeared corresponding to the $\beta$-CH$_2$ protons of GSSG (Fig. 2c). Using high-resolution electron spray ionization mass spectrometry (HR-ESI-MS), the presence of GSSG ([M-H]$^-$ = 611.1442) was verified (Supplementary Fig. 11). For a quantitative insight, the solution of Ru(II)-OH and GSH

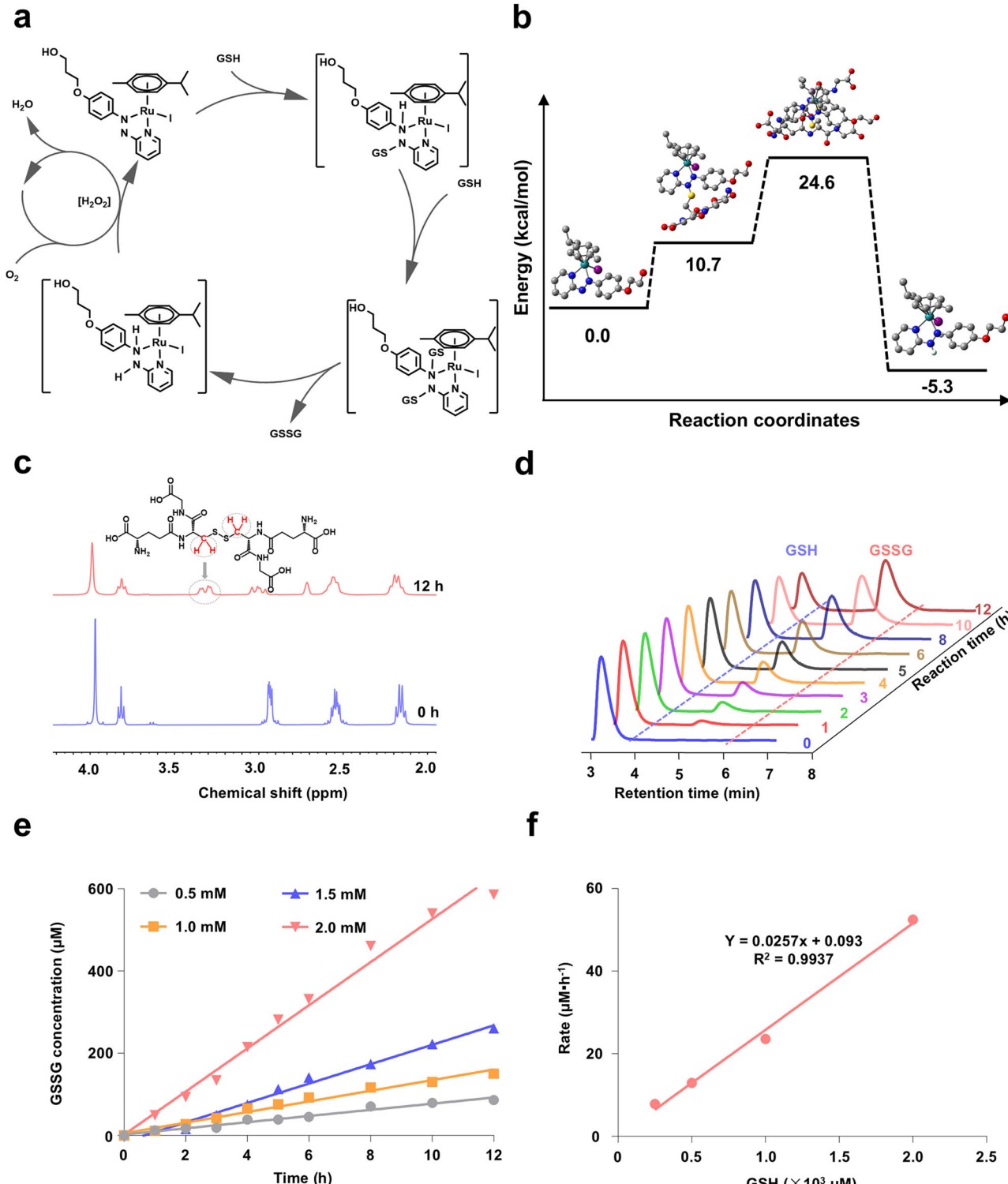

**Fig. 2 | Density functional theory calculations and characterization of the catalytic properties of Ru(II)-OH. a** Proposed catalytic cycle of the transformation of GSH into glutathione disulfide (GSSG) using the ruthenium (Ru) catalyst Ru(II)-OH. **b** Energy profile diagram of the respective energetic states in the catalytic transformation. The presence of GSH, GSSG, and hydrogen atoms were considered in the calculation of the energetic states but in this schematic representation omitted for clarity. **c** ¹H-NMR spectra of GSH (50 mM) in the presence of Ru(II)-OH (2 mM) directly after preparation (blue) and after incubation for 12 h (red). **d** Ultra performance liquid chromatograms (UPLC) of the conversion of GSH (2 mM) to GSSG in the presence of Ru(II)-OH (50 µM). **e** Conversion of various concentrations of GSH to GSSG in the presence of Ru(II)-OH in dependence of the reaction time. **f** Correlation of reaction rate in dependence of the GSH concentration of the conversion of GSH to GSSG in the presence of Ru(II)-OH.

was time-dependent monitored by ultra performance liquid chromatography (UPLC). While the peak for GSH (2.98 min) gradually decreased, the signal corresponding to GSSG increased (4.37 min), indicative of the expected chemical transformation (Fig. 2d). The incubation of solutions of GSH without the metal complex did not show any conversion to GSSG (Supplementary Fig. 12). Importantly, the retention time and integral of the peak corresponding to Ru(II)-OH remained unchanged, suggestive that this is a catalytic process (Supplementary Fig. 13). Using different concentrations of GSH (0.5–2 mM), the reaction kinetics were studied. The concentration of the product increased proportionally with the incubation time (chromatograms: Supplementary Fig. 14, quantification: Fig. 2e). The reaction rate was found to be in a linear correlation with the GSH concentration (Fig. 2f), indicative of pseudo first order reaction kinetics. The analysis of the catalytic parameters demonstrated a reaction rate of $0.0257\,h^{-1}$, turnover number of 38.2, and turnover frequency of Ru(II)-OH of $0.3141\,h^{-1}$ (Supplementary Table 1). Combined these findings suggest that Ru(II)-OH could efficiently and catalytically oxidize GSH to GSSG.

## Encapsulation into nanoparticles

To enhance the pharmacological properties of the metal complex and provide cancer selectivity, Ru(II)-OH was incorporated into a polymer backbone that could self-assemble into tumor-targeting nanoparticles. Mercaptoacetic acid was dimerized in the presence of trifluoroacetic acid and acetone and then reduced with lithium aluminum hydride into the linker 2,2′-(propane-2,2-diylbis(sulfanediyl))diacetic acid (synthetic strategy: Supplementary Fig. 2). The compounds were characterized by NMR spectroscopy (Supplementary Fig. 15). The polymer P3 was synthesized by condensation polymerization using 2,2′-(propane-2,2-diylbis(sulfanediyl))bis(ethan-1-ol), 1H,1H,10H,10H-perfluoro-1,10-decanediol, and 1,2,4,5-cyclohexanetetracarboxylic acid dianhydride (Supplementary Fig. 4). Using $^{1}$H- and $^{19}$F-NMR spectroscopy as well as gel permeation chromatography (Supplementary Figs. 18–21), the average molecular weight was determined to be approximately 22 kDa with a polydispersity index (PDI) of 1.08. The polymeric material P3 was posttranslational functionalized with the ruthenium catalyst through ester bond formation to generate P4. Inductive coupled plasma-mass spectrometry (ICP-MS) measurements indicated a loading of ~22%. Analogously, polymers without polyfluorinated moieties, followingly referred to as P1 and the ruthenium catalyst derivative P2 were prepared (Supplementary Fig. 3, Supplementary Figs. 16–17). P1 was found with an approximate size of 21 kDa and a PDI of 1.10. P2 had a drug loading of approximately 23%. Based on the amphiphilic nature of the polymers, these could self-assemble into nanoparticles within an aqueous solution. Followingly, the nanoparticles generated from P4 are referred to as NP4 and from P2 as NP2 (Fig. 3a). Dynamic light scattering (DLS) measurements demonstrated that NP4 had an average hydrodynamic diameter of 126 nm and a PDI of 0.16 (Fig. 3b) and NP2 had an average hydrodynamic diameter of 116 nm and a PDI of 0.07 (Supplementary Fig. 22). NP2 had a negative zeta potential of −8.88 mV and NP4 of −3.77 mV (Fig. 3b). Transmission electron microscopy (TEM) images showed the spherical morphology of the nanoparticles (NP2: Supplementary Fig. 23, NP4: Fig. 3c). Scanning transmission electron microscopy coupled with energy dispersive X-ray spectroscopy images indicated the presence and uniform distribution of sulfur and ruthenium for NP2 (Supplementary Fig. 24) and fluorine, sulfur, and ruthenium for NP4 (Fig. 3d). These results indicate the successful incorporation of the ruthenium catalyst into polymers that could self-assemble into well dispersed nanoparticles. Meanwhile, the stability of nanoparticles was evaluated using DLS. The results showed that the average hydrodynamic diameter and PDI of NP4 remained unchanged over a 14-day period, indicative of the excellent stability of NP4 (Supplementary Fig. 25).

Nanoparticles were designed with a 2,2′-(propane-2,2-diylbis(sulfanediyl)) bis(ethan-1-ol) linker that could be oxidized by ROS, and

decompose into oligomeric units, and release the ruthenium catalyst. To investigate this potential release, NP4 was incubated with phosphate-buffered saline (PBS) or PBS containing $H_2O_2$ as a model for ROS inside the cancer cells and the release monitored by ICP-MS. While NP4 remained relatively stable under physiological conditions, the payload was readily released in the presence of $H_2O_2$ (Supplementary Fig. 26). The ability of the nanoparticles to store and transport oxygen was studied using a dissolved oxygen analyzer. Oxygen gas was bubbled through aqueous solutions of NP2 and NP4 and the amount of trapped oxygen time-dependently quantified. NP4 reached a maximal oxygen storage of 53 mg/L, which was approximately 2.5 more than for NP2 (Supplementary Fig. 27), indicative of the oxygen store and transport properties of the carbon fluorinated nanoparticles NP4.

## Therapeutic effects in hypoxic osteosarcoma cells

The biological activities of NP4 were studied against human osteosarcoma (143B) and mouse osteosarcoma (K7M2) cells with different $O_2$ concentrations. As a model for physiological conditions the cancer cells were cultivated and treated with a 21% $O_2$ atmosphere, and as a model for hypoxic conditions the cancer cells were cultivated and treated with a 1% $O_2$ atmosphere. It is important to mention that these $O_2$ levels do not represent the concentrations found inside human healthy and cancerous tissues and are followingly used to investigate the different therapeutic efficiency of the nanoparticle formulation in high and low $O_2$ atmospheres. The cellular uptake of the nanoparticles in 143B cells was investigated upon labeling of NP4 with the well-characterized dye Cy5.5 into the nanoparticle formulation NP4-Cy5.5. The time-dependent monitoring by confocal laser scanning microscopy (CLSM) indicated that with longer incubation time, an increasing fluorescence intensity of NP4-Cy5.5 was detected in the cancer cells (Fig. 4a). Complementary, this finding was confirmed by flow cytometry (flow cytometry plots: Fig. 4b, quantification: Fig. 4c). Cellular uptake studies were performed in cellular environments with different oxygen concentrations in the cell lines 143B and K7M2. The results showed no significant dependence of the cellular uptake of nanoparticles on the oxygen concentration in the cells (Supplementary Fig. 28). The biodegradability of NP4 upon internalization into the cancer cells was studied upon encapsulation of the dye Nile Red into a nanoparticle formulation, followingly referred to as NP4-NR. When Nile Red is in the hydrophobic core of the nanoparticles, it is highly fluorescent and can be easily detected. However, upon release in the hydrophilic cytoplasm, the fluorescence of the dye is quenched, allowing for the facile detection of the degradation of the nanoparticles. CLSM images upon incubation of NP4-NR with the cancer cells showed the steadily increase of the fluorescence inside the cancer cells, reaching a maximal emission after incubation for 4 h. Further monitoring of the cancer cells for an additional 10 h demonstrated the gradual decrease of the red fluorescence signal, indicative of the degradation of the nanoparticles inside the cancer cells (Supplementary Fig. 29).

The cytotoxicity of Ru(II)-OH, NP2, and NP4 against 143B and K7M2 cells in a 21% $O_2$ or 1% $O_2$ atmosphere was studied using a 3-(4,5-dimethylthiazol-2-yl)-2,5-diphenyltetrazolium bromide (MTT) assay. The therapeutic effects of the compounds were found to be comparable between 143B and K7M2 cells. Interestingly, in a 21% $O_2$ atmosphere the cytotoxic effect of Ru(II)-OH, NP2, and NP4 was found to be in the same range ($IC_{50} = 6.1$–10.4 μM). Strikingly, the treatment under 1% $O_2$ conditions showed drastic differences. Ru(II)-OH and NP2 were found to be non-toxic ($IC_{50} > 15$ μM) in both cell lines under 1% $O_2$ conditions. Contrary, NP4 showed a cytotoxic effect in the very low micromolar range (143B: $IC_{50} = 6.6$ μM, K7M2: $IC_{50} = 2.1$ μM) (drug-response curves: Fig. 4d, overview of $IC_{50}$ values: Supplementary Table 2). Using a cell live (calcein AM) and cell death (propidium iodide) stain, the cytotoxic effect in 143B cells at a fixed concentration of 10 μM Ru was visualized. The cells treated with Ru(II)-OH, NP2, or NP4 under 21% $O_2$ conditions showed a reduced cell population as well

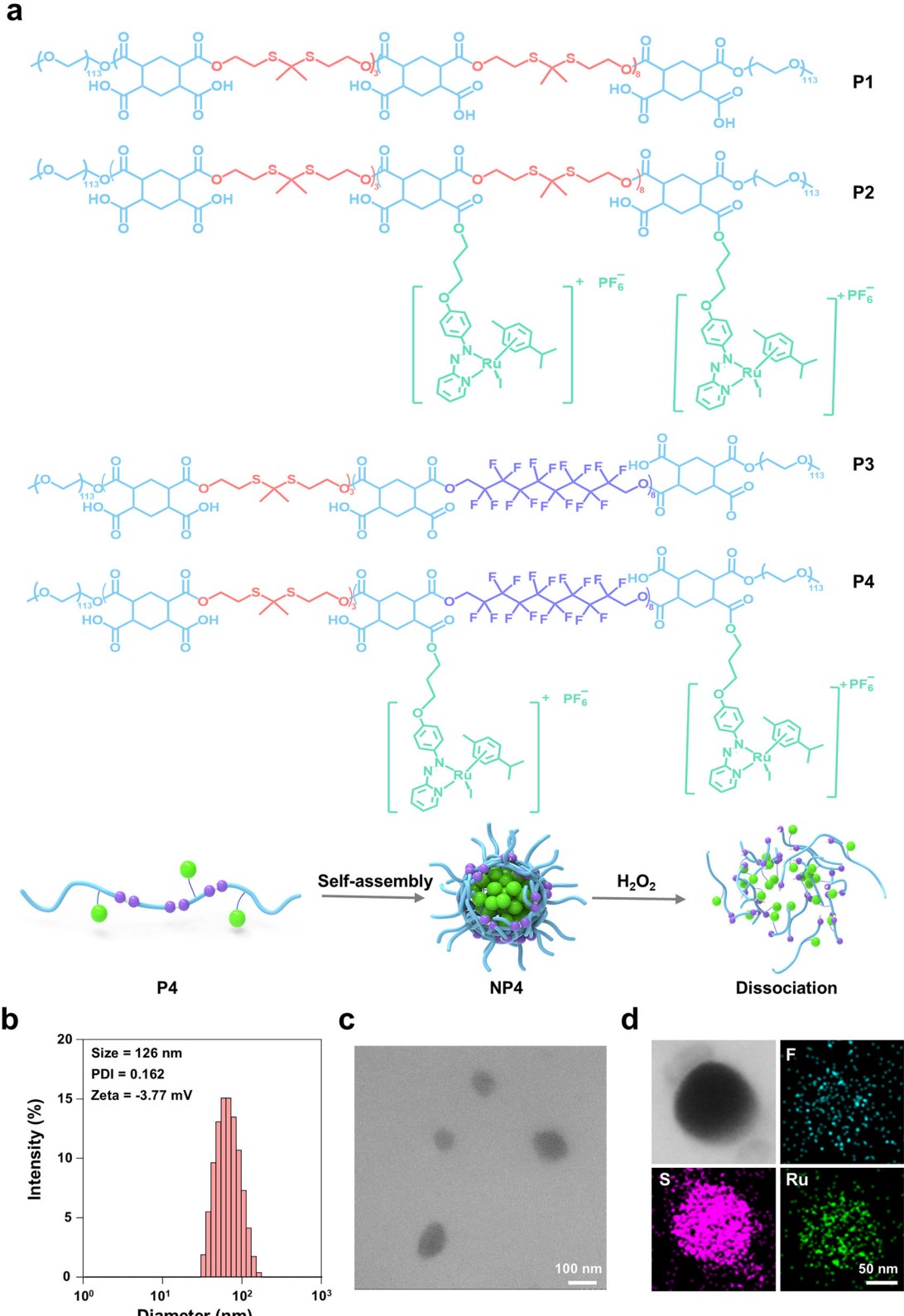

**Fig. 3 | Preparation and characterization of nanoparticles. a** Structural formula of polymers (P1-4), and schematic illustration of the self-assembly and $H_2O_2$ induced decomposition of NP4. **b** Hydrodynamic diameter of NP4 determined by dynamic light scattering (DLS) measurements. **c** Representative transmission electron microscopy (TEM) image of NP4. **d** Representative transmission electron microscopy energy-dispersive X-ray spectroscopy mapping of NP4.

as a mixture of living and dead cells comparable between the respective treatment. While the cells treated with Ru(II)-OH or NP2 in a 1% $O_2$ atmosphere were confluent and alive, the cell treated with NP4 had a strongly reduced cell population and a mixture of living and dead cells comparable to the treatment with these nanoparticles in a 21% $O_2$ atmosphere (Fig. 4e). The ability to induce apoptosis in a 1% $O_2$ atmosphere was studied using an Annexin V-FITC and propidium iodide dual stain. While only minimal amount of apoptosis upon

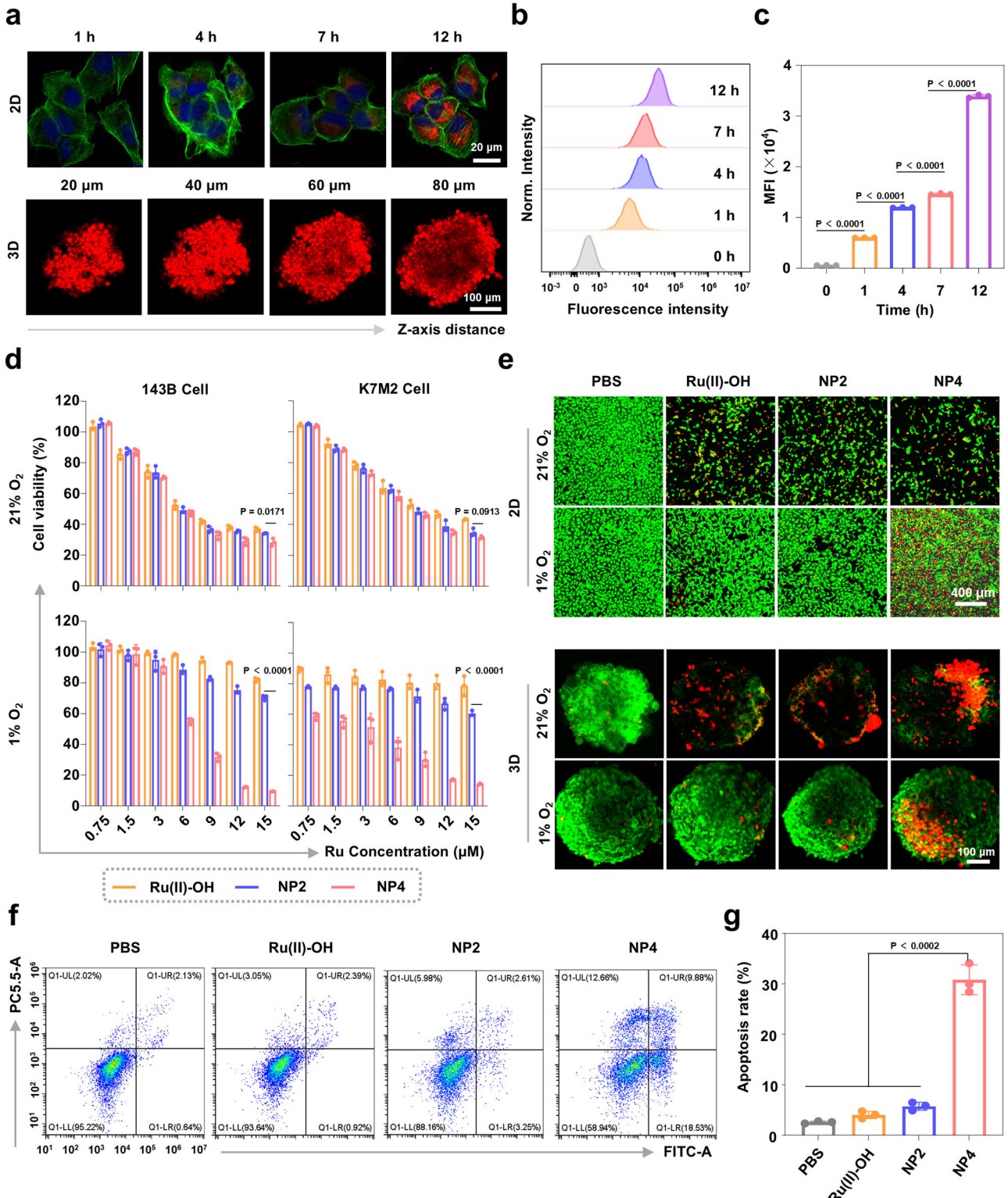

**Fig. 4 | Cellular uptake and cytotoxic effects of Ru(II)-OH, NP2, and NP4 against 143B and K7M2 cells under 21% O₂ and 1% O₂ conditions. a** Time-dependent confocal laser scanning microscopy (CLSM) images of the cellular uptake of NP4-Cy5.5 in 143B cells or 143B multicellular tumor spheroids. The cell nucleus was stained with 4′, 6-diamidino-2-phenylindol (DAPI), and the cell cytoskeleton was stained with Alexa-488. **b** Time-dependent cellular uptake of NP4-Cy5.5 in 143B cells determined by flow cytometry. **c** Quantification of the cellular uptake of NP4-Cy5.5 from (**b**) (n = 3 independent experiments). **d** Representative drug response curves of Ru(II)-OH, NP2, and NP4 against 143B and K7M2 cells in a 1% or 21% O₂ atmosphere,

(n = 3 independent experiments). **e** Representative CLSM images of 143B cells and 143B multicellular tumor spheroids under 21% O₂ or 1% O₂ conditions treated with of Ru(II)-OH, NP2, and NP4 (10 μM Ru) for 24 h and stained with cell live (calcein AM, green) and cell death (propidium iodide, red). **f** Flow cytometry plots of 143B cells under 1% O₂ conditions treated with of Ru(II)-OH, NP2, and NP4 (10 μM Ru) for 12 h and stained with Annexin V-FITC and propidium iodide. **g** Determination of the apoptosis rate from (**f**) (n = 3 independent experiments). Data are presented as mean ± standard deviation (SD). Statistical significance between every two groups was calculated by T-test, the statistical test used was two-sided.

treatment with Ru(II)-OH or NP2 was observed, the treatment with NP4 trigger high amounts of early and late apoptotic processes (flow cytometry plots: Fig. 4f, quantification: Fig. 4g). Interestingly, the incubation with the antioxidant N-acetylcysteine (NAC) upon treatment with NP4 resulted in reduced levels of apoptosis, suggestive that the cytotoxicity of NP4 is related to intracellular redox homeostasis (Supplementary Fig. 30). Overall, these findings collectively demonstrate that the molecular ruthenium catalyst Ru(II)-OH and the nanoparticle formulation NP2 are therapeutically active in a 21% $O_2$ atmosphere, but these lose their therapeutic efficiency in a 1% $O_2$ atmosphere. In contrast, NP4 remained cytotoxic in a 1% and 21% $O_2$ atmosphere.

Following the evaluation inside two-dimensional monolayer cancer cells, the therapeutic properties were studied in three-dimensional multicellular tumor spheroids. A multicellular tumor spheroid is a tissue culture model that mimics the pathological conditions of solid tumors including a three-dimensional cellular structure, nutrition, and proliferation gradients. The penetration of the multicellular tumor spheroid was studied by z-stack CLSM using NP4-Cy5.5. An emission signal was observed at every section depth, suggestive of the full penetration of the three-dimensional structure (Fig. 4a). To investigate the ability to generate a therapeutic effect, the multicellular tumor spheroids were treated with Ru(II)-OH, NP2, or NP4 (10 µM Ru) in a 1% or 21% $O_2$ atmosphere and afterward incubated with a cell live or death stain. The spheroids treated with Ru(II)-OH, NP2, or NP4 in a 21% $O_2$ atmosphere primary consisted of dead cells. While the spheroids treated with Ru(II)-OH or NP2 in a 1% $O_2$ atmosphere majorly consisted of living cells, the spheroids treated with NP4 in a 1% $O_2$ atmosphere consisted of dead cells (Fig. 4e). These results indicate that NP4 are able to fully penetrate the three-dimensional cellular structure and generate a cytotoxic effect in multicellular tumor spheroids.

## GSH oxidation catalysis in hypoxic osteosarcoma cells

Based on these promising therapeutic properties in a 1% $O_2$ atmosphere, the biological mechanism of action was in-depth studied. The ability to catalytically convert GSH into GSSG inside the cancer cells was assessed upon determination of the levels of GSH and GSSG after treatment. The cancer cells that were incubated with PBS were found with a GSH/GSSG ratio of 1.55. In contrast, upon treatment with NP4 in a 1% $O_2$ atmosphere, the GSH/GSSG ratio dropped to 0.12. Upon preincubation of the cancer cells with the antioxidant NAC and subsequent treatment with NP4, the GSH/GSSG ratio was significantly less influenced, reaching a GSH/GSSG ratio of 0.71 (Fig. 5a), suggestive of the importance of the redox hemostasis for the depletion of GSH from the cancer cells. As GSH represent the primary antioxidant defense system, the reduction of GSH could influence the redox homeostasis and elevate the naturally occurring ROS levels inside the cancer cells[40]. Consequently, changes in the ROS levels inside the cancer cells in a 1% $O_2$ atmosphere were examined by utilizing the ROS specific probe 2′,7′-dichlorodihydrofluorescein diacetate (DCFH-DA) which is chemically converted into a green fluorescent dye in the action of ROS. While the treatment with Ru(II)-OH or NP2 slightly elevated the ROS levels inside the hypoxic cancer cells, the treatment with NP4 showed a strong enhancement of ROS. For a deeper understanding, the cancer cells were preincubated with NAC and the presence of ROS upon treatment were assessed using the ROS specific probe DCFH-DA. While the treatment with NP4 alone caused a strong production of ROS, the preincubation with NAC and subsequent treatment with NP4 showed a poor ROS generation (Fig. 5d). Following the semi qualitative assessment, the ROS levels were quantified by flow cytometry. The results demonstrated a very strong accumulation of ROS inside the hypoxic cancer cells upon treatment with NP4 but a poor ROS accumulation upon preincubation with NAC and subsequent treatment with NP4 (flow cytometry plots: Fig. 5b, quantification: Fig. 5c). Based on the disruption of the

antioxidant defense system and the accumulation of ROS, lipid peroxide could be formed inside the cancer cell[41]. To investigate this, the cancer cells were treated with the compounds and afterwards stained with the lipid peroxide specific probe C11-BODIPY. CLSM images showed the presence of lipid peroxides throughout the cancer cells upon treatment with Ru(II)-OH, NP2, or NP4 (Fig. 5e). A comparison showed that the treatment with NP4 resulted in the highest amount of lipid peroxides (Supplementary Fig. 31). For a quantitative insight, this finding was further confirmed by flow cytometry (flow cytometry plots: Fig. 5f, quantification: Fig. 5g). Within the proposed catalytic cycle (Fig. 2), the consumption of $O_2$ for the regeneration of the ruthenium catalyst is proposed.

To investigate the depletion of $O_2$ within the cells, cancer cells in a 1% $O_2$ atmosphere were treated with Ru(II)-OH, NP2, or NP4, and the $O_2$ concentration was assessed using the hypoxia probe pimonidazole hydrochloride. In the presence of $O_2$, the fluorescence of the probe is quenched and no fluorescence is measured. Contrary, in hypoxic regions a strong green emission is observed. The treatment with Ru(II)-OH or NP2 showed a strong green fluorescence signal throughout the cell, indicative of the depletion of $O_2$ and therefore the $O_2$ consumption through the catalyst. Contrary, the untreated cells in a 1% $O_2$ atmosphere and the cells treated with NP4 did not show any emission, suggestive of the presence of $O_2$ (CLSM images: Fig. 5h, quantification: Supplementary Fig. 32). In contrast to Ru(II)-OH or NP2, the nanoparticle formulation NP4 has a carbon fluorinated moiety in the polymeric backbone, which could be used for the transport of $O_2$ and therefore explain this phenomenon.

## Macrophage activation in hypoxic osteosarcoma cells

$H_2O_2$ is one of the main products of the catalytic reaction. Therefore, the intracellular levels of $H_2O_2$ under 1% $O_2$ conditions were further evaluated through CLSM and assay kits. The CLSM images showed that the intracellular levels of $H_2O_2$ were higher in cells treated with NP4 compared to cells treated with Ru(II)-OH and NP2 (Fig. 6a), with a concentration of approximately 1.5 times higher than cells treated with Ru(II)-OH and NP2 (Fig. 6b). $H_2O_2$, as a relatively stable ROS, can diffuse across cell membranes and cellular compartments, acting as an important second messenger molecule involved in regulating various physiological processes such as cell proliferation, differentiation, energy metabolism, and immune responses. Therefore, the concentration of $H_2O_2$ in the cell supernatant was further detected using assay kits. The results showed that the concentration of $H_2O_2$ in the cell supernatant treated with NP4 was higher compared to the supernatant of cells treated with Ru(II)-OH and NP2 (Fig. 6c), indicating that $H_2O_2$ catalyzed by NP4 not only disrupted the redox balance within tumor cells but also diffused outside the cells, leading to more cascading reactions. According to the literature, $H_2O_2$ can activate the TRPM2 ion channels in macrophages, promoting calcium influx and mediating immune responses[42–44]. The supernatant of tumor cells treated with Ru(II)-OH, NP2, and NP4 was co-cultured with RAW264.7 macrophages, and the intracellular calcium levels in macrophages were evaluated (Fig. 6d). CLSM results indicated that the supernatant of tumor cells incubated with NP4 effectively promoted calcium influx in macrophages. Furthermore, Western blot results also indicated that the supernatant of tumor cells treated with NP4 promoted TRPM2 protein expression in macrophages (Fig. 6e). The secretion of immune cytokines primarily manifests the activation of macrophages. Therefore, the levels of cytokines (IL-6, IL-12, TNF-α) in the macrophage supernatant were detected using an ELISA assay (Fig. 6f-h). The results revealed that the levels of pro-inflammatory cytokines IL-6, IL-12, and TNF-α secreted by cells treated with NP4 were 1.45, 5, and 1.78 times higher than those secreted by cells treated with NP2, respectively, further confirming the activation of macrophages. The cancer cells were preincubated with NAC and subsequently treated with NP4 showed a reduced number of TNF-α secreted from the macrophages, suggesting that macrophage

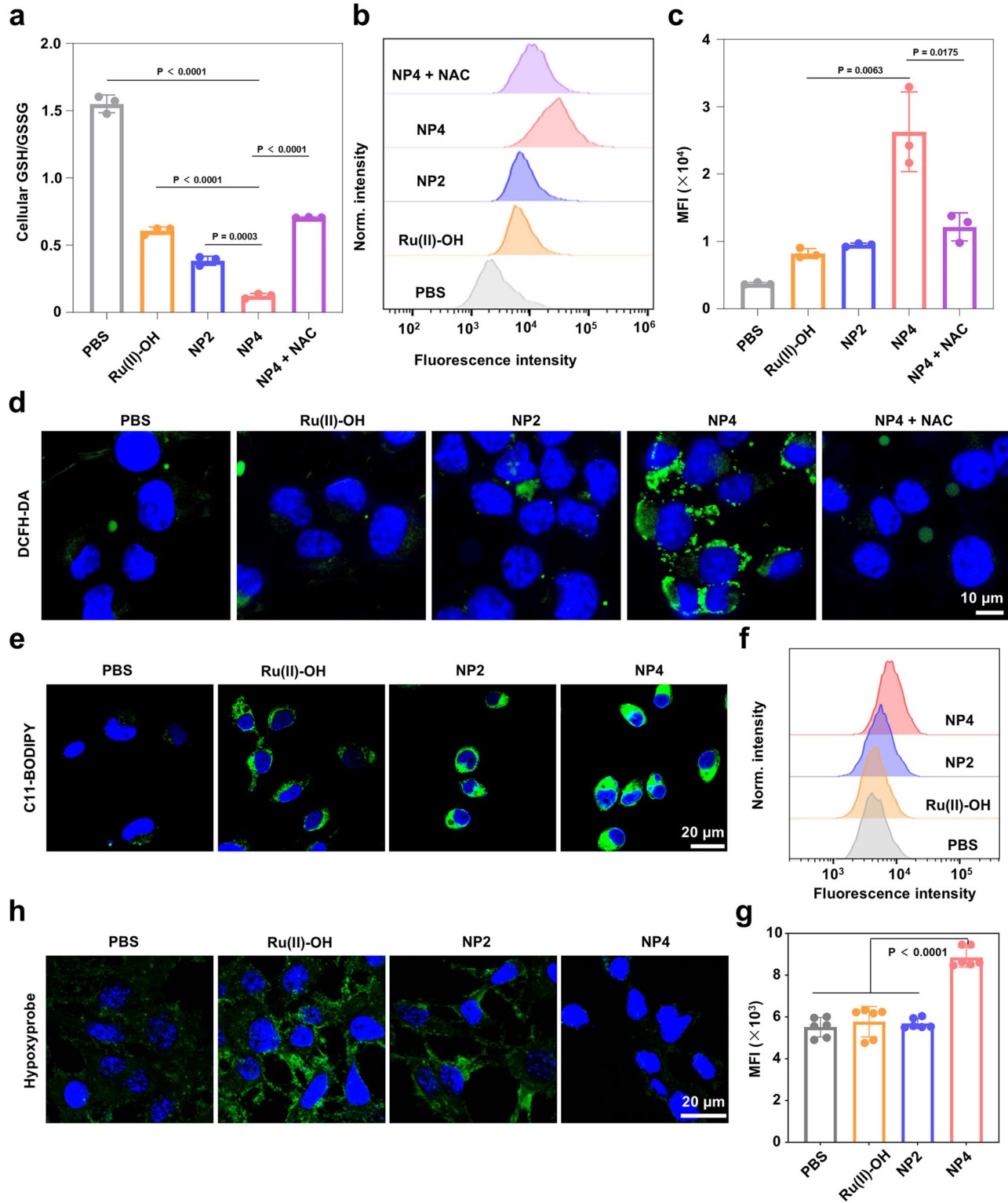

activation is associated with the NP4-induced oxidative stress. To validate macrophage activation, mouse bone marrow-derived macrophages were extracted and incubated with cancer cells, that have been treated with the nanoparticles. The macrophage activation was analyzed by flow cytometry. The results showed that the treated cancer cells promoted the differentiation of the mouse bone marrow-derived macrophages. The treatment with NP4 enhanced the level of pro-inflammatory M1 macrophages by 97% in comparison to the treatment with NP2. In addition, the treatment with NP4 reduced the level of anti-inflammatory M2-type macrophages by 20% in comparison to the

treatment with NP2 (Supplementary Fig. 33). Combined these results indicated that NP4 catalyzes the generation of $H_2O_2$ in tumor cells, which diffuses outside the cells and activates macrophages, thereby triggering immune responses.

## Biodistribution and therapeutic effects in orthotopic osteo-sarcoma mouse model

Based on these promising findings, the biosafety of Ru(II)-OH, NP2, or NP4 at a doses of 2 mg Ru/kg was studied upon intravenous injection into the tail vein of healthy female mice on the days 1, 4, and 7. The

**Fig. 5 | Consumption of O₂, accumulation of ROS, conversion of GSH, and accumulation of lipid peroxides in 143B cells under 1% O₂ conditions upon treatment with Ru(II)-OH, NP2, NP4 or combined NP4 with N-acetylcysteine (NP4 + NAC). a** GSH/GSSG ratio inside the cancer cells upon various treatments, n = 3 independent experiments. **b** Flow cytometry plots of 143B cells in a 1% O₂ atmosphere treated with Ru(II)-OH, NP2, NP4, and NP4 + NAC (10 μM Ru) for 6 h and stained with 2′,7′-dichlorodihydrofluorescein diacetate (DAFH-DA). **c** Quantification from (**b**), n = 3 independent experiments. **d** Representative CLSM images of 143B cells in a 1% O₂ atmosphere treated with of Ru(II)-OH, NP2, NP4, and NP4 + NAC (10 μM Ru) for 6 h and stained with the ROS probe 2′,7′-dichlorodihydrofluorescein diacetate (DCFH-DA, green) and DAPI (blue, nucleus).

**e** Representative CLSM images of 143B cells in a 1% O₂ atmosphere treated with of Ru(II)-OH, NP2, and NP4 (10 μM Ru) for 6 h and stained with the lipid peroxide specific probe C11-BODIPY (green) and DAPI (blue, nucleus). **f** Flow cytometry plots of 143B cells in a 1% O₂ atmosphere treated with of Ru(II)-OH, NP2, and NP4 (10 μM Ru) for 6 h and stained with the lipid peroxide specific probe C11-BODIPY. **g** Quantification from (**f**), n = 6 independent experiments. **h** Representative CLSM images of 143B cells under 1% O₂ conditions treated with of Ru(II)-OH, NP2, and NP4 (10 μM Ru) for 12 h and stained with the hypoxia probe pimonidazole hydrochloride (green, hypoxia areas) and DAPI (blue, nucleus). Data are presented as mean ± SD. Statistical significance between every two groups was calculated by T-test, the statistical test used was two-sided.

animal models behave normally without signs for pain, stress, or discomfort. After 14 days, the mice were sacrificed, and their major organs and tissues were analyzed with a hematoxylin and eosin (H&E) stain (Supplementary Fig. 34). The results showed no histological alterations, indicative of the high biocompatibility and biosafety. The blood circulation of the nanoparticles was assessed upon intravenous injection into the tail vein of healthy female mice and determination of the metal content in the blood by ICP-MS. As expected, the molecular metal complex Ru(II)-OH was quickly cleared from the blood stream with a blood circulation half-life time of 0.28 h. In contrast, the nanoparticles NP4 had a much longer blood circulation half-life time of 2.97 h (Supplementary Fig. 35), suggesting a higher bioavailability of NP4 in comparison to Ru(II)-OH.

Next, an orthotopic K7M2 osteosarcoma model with an active immune system was established to investigate the tumor targeting and therapeutic effects of NP4 (Fig. 7a). To visualize the biodistribution of nanoparticles in mice, NP4 was labeled with the fluorescence dye Cy7.5, resulting in NP4-Cy7.5. Subsequently, NP4-Cy7.5 was intravenously injected into BALB/c mice, and the biodistribution of nanoparticles in mice was observed using an in vivo imaging system (IVIS). The results showed that the fluorescence intensity at the tumor site gradually increased over time, reaching a peak at 6 h and maintaining a relatively high intensity for the subsequent 42 h, indicating the rapid and prolonged accumulation of nanoparticles at the tumor site (Fig. 7b, c). After 48 h, the mice were sacrificed, and the tumors as well as major organs (heart, liver, spleen, lung, kidney) were collected to measure their fluorescence intensity. The results revealed that, except for the metabolic organs liver and kidney, the tumor tissue exhibited the highest fluorescence intensity, further confirming the excellent tumor targeting of NP4 (Fig. 7d). Complementary, the high tumor accumulation was confirmed upon determination of the metal content in the respective organs by ICP-MS. The direct comparison upon using the same concentration of ruthenium of the molecular agent Ru(II)-OH and NP4 showed a significantly higher tumor accumulation of the nanoparticle formulation NP4 (Supplementary Fig. 36). The remarkable tumor-targeting ability of NP4 endows it with significant potential for tumor therapy.

K7M2-Luciferase (K7M2-Luc) cells are cancer cells that express the reporter gene luciferase, which reacts with its substrate (luciferin) to generate chemiluminescence. Therefore, bioluminescence imaging (BLI) can be employed to non-invasively and accurately detect the size of live tumors. Thus, the therapeutic effect of NP4 on the orthotopic K7M2-Luc osteosarcoma mouse model was evaluated (Fig. 7e, quantification: Supplementary Fig. 37). The results showed that the chemiluminescence intensity at the tumor site gradually weakened or even disappeared with the progression of NP4 treatment, while the chemiluminescence intensity in mice without drug treatment and those treated with Ru(II)-OH and NP2 gradually increased. The imaging results demonstrated that NP4 effectively inhibited the growth of orthotopic osteosarcoma. During the treatment, no significant change in the body weight of the mice was observed, indicating that NP4 has a high biocompatibility (Supplementary Fig. 38). After 12 days, the mice were sacrificed, and the tumor tissues were collected. The conclusion

was further validated by terminal deoxynucleotidyl transferase dUTP nick end labeling (TUNEL) and H&E staining (Fig. 7f, Supplementary Fig. 39), which revealed that the tumor tissues of mice treated with NP4 exhibited the strongest green fluorescence (apoptotic cells), demonstrating the good anti-tumor activity of NP4.

## Immune activation in orthotopic osteosarcoma mouse model

Cell experiments demonstrated that H₂O₂ generated by NP4 could activate macrophages and mediate immune responses. Next, the effectiveness of NP4 in activating the immune response in mice in orthotopic K7M2 osteosarcoma models was further investigated. (Fig. 8a). Tumor-draining lymph nodes (TDLNs) and tumor tissues from mice upon various treatments were collected to study the changes in relevant immune indicators. The maturation of dendritic cells (DCs) in the TDLNs and tumors of mice upon various treatments was first analyzed using flow cytometry. The results indicated that the proportion of mature DCs (CD80⁺ CD86⁺) in the TDLNs of mice treated with NP4 (82.8%) was approximately 1.46 times higher than that in the mice treated with NP2 (56.7%) (Fig. 8b, c). Additionally, compared to the tumors of mice without drug treatment and those treated with Ru(II)-OH (64.4%) and NP2 (73.6%), the tumors of mice treated with NP4 (81.2%) exhibited a higher proportion of mature DCs (CD80⁺ CD86⁺) (Supplementary Fig. 40). These experimental results fully demonstrate that NP4 can promote the maturation of DCs.

Mature DCs can elicit efficient adaptive immune responses through the activation of T lymphocytes. Therefore, the infiltration of helper T cells (CD4⁺) and cytotoxic T cells (CD8⁺) in tumor tissues was further investigated. It was found that the proportion of CD4⁺ T cells in mouse tumor tissues increased from 42.1% to 55.5% after NP4 treatment, while the proportion of CD8⁺ T cells increased from 19.0% to 28.9% (Fig. 8d, e). Tumor-associated macrophages (TAMs), which are infiltrating macrophages in tumor tissues, can be divided into M1-type macrophages that inhibit tumor growth and M2-type macrophages that promote tumor growth. In this study, the effect of NP4 on macrophages in tumor tissues was evaluated using flow cytometry. It was showed that NP4 treatment leaded to an increase in pro-inflammatory M1-type macrophages and a decrease in anti-inflammatory M2-type macrophages in mouse tumor tissues, with the proportion of M1 to M2 macrophages being 4.4 times higher compared to untreated mouse tumor tissues (Fig. 8f, g). This suggests that NP4 could effectively promote the polarization of M2-type macrophages towards M1-type macrophages, thereby alleviating immune suppression. In addition, immunohistochemical analysis of tumor tissue sections also demonstrated that NP4 could increase the infiltration of CD8⁺ T cells (Fig. 8h). These experimental results demonstrated that NP4 could activate macrophages, facilitate DCs maturation, increase infiltration of cytotoxic T cells, activate the mouse immune system, and inhibit tumor growth in osteosarcoma.

## Biodistribution and therapeutic effects in subcutaneous osteosarcoma mouse model

The therapeutic properties of the nanoparticles were further studied inside an 143B tumor-bearing immunodeficient nude mouse model

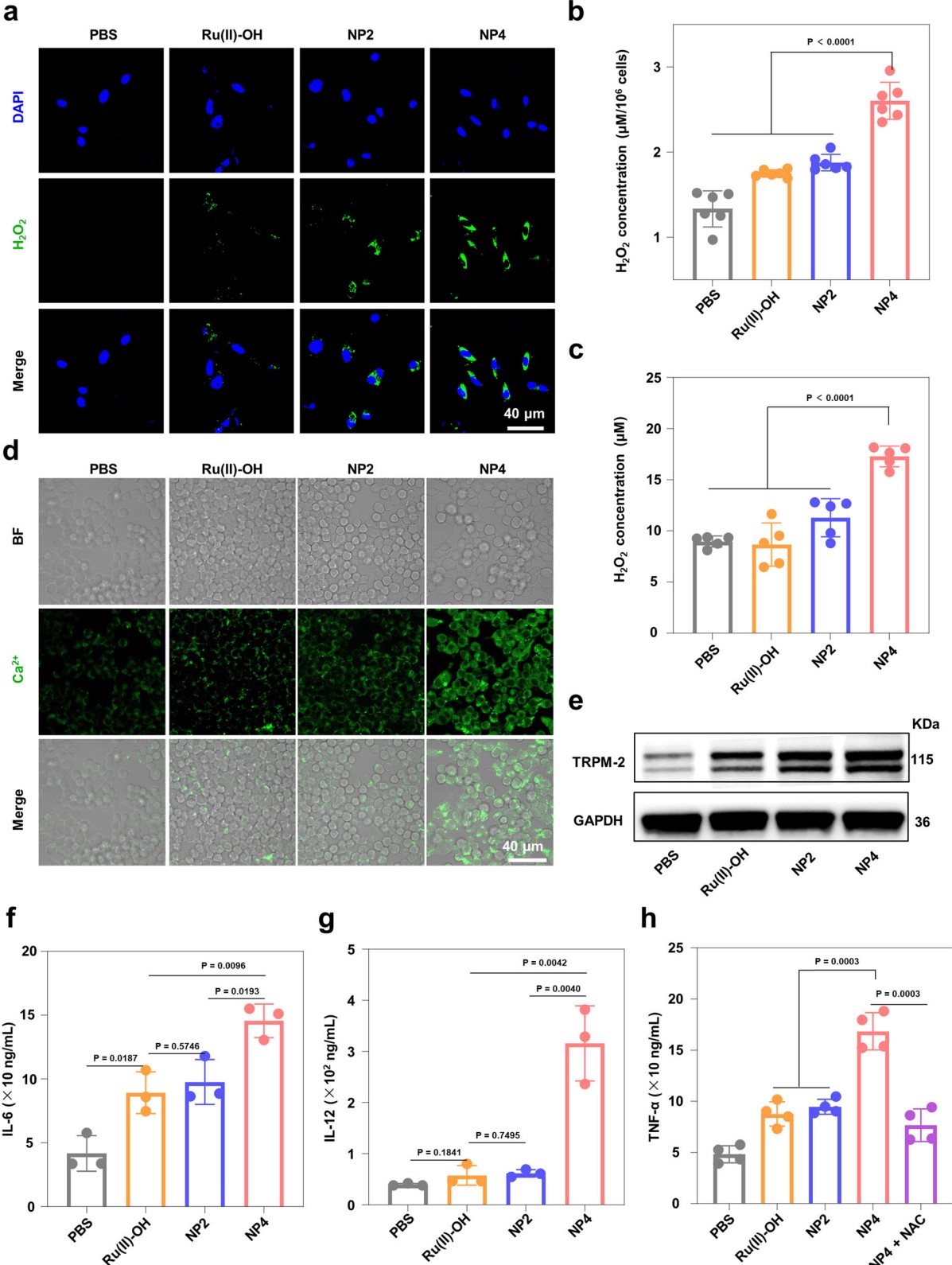

**Fig. 6 | The macrophage activation by NP4. a** Representative CLSM images of 143B cells under 1% $O_2$ conditions treated with of Ru(II)-OH, NP2, and NP4 (10 μM Ru) for 6 h and stained with the ROS Green™ $H_2O_2$ Probe (green, $H_2O_2$) and DAPI (blue, nucleus). **b** $H_2O_2$ concentration inside the cancer cells upon various treatments (PBS, Ru(II)-OH, NP2, and NP4 (10 μM Ru)), n = 6 independent experiments. **c** $H_2O_2$ concentration in cancer cell supernatants upon various treatments (PBS, Ru(II)-OH, NP2, and NP4 (10 μM Ru)), n = 5 independent experiments. **d** Representative CLSM images of RAW264.7 macrophage in a 1% $O_2$ atmosphere treated with of Ru(II)-OH, NP2, and NP4 (10 μM Ru) for 6 h and stained with the Calcium ionization probe Fluo-4 AM (green, $Ca^{2+}$). **e** Western blot analysis for the expression of TRPM2 proteins in RAW264.7 macrophage cells treated with PBS, Ru(II)-OH, NP2, and NP4 for 12 h. Concentration of IL-6 (**f**), IL-12 (**g**), and TNF-α (**h**) in RAW264.7 macrophage supernatants determined by ELISA assay upon various treatments (PBS, Ru(II)-OH, NP2, NP4, or NP4 + NAC (10 μM Ru)). For (**f**, **g**) n = 3 independent experiments, for (h) n = 4 independent experiments. Data are presented as mean ± SD. Statistical significance between every two groups was calculated by T-test, the statistical test used was two-sided.

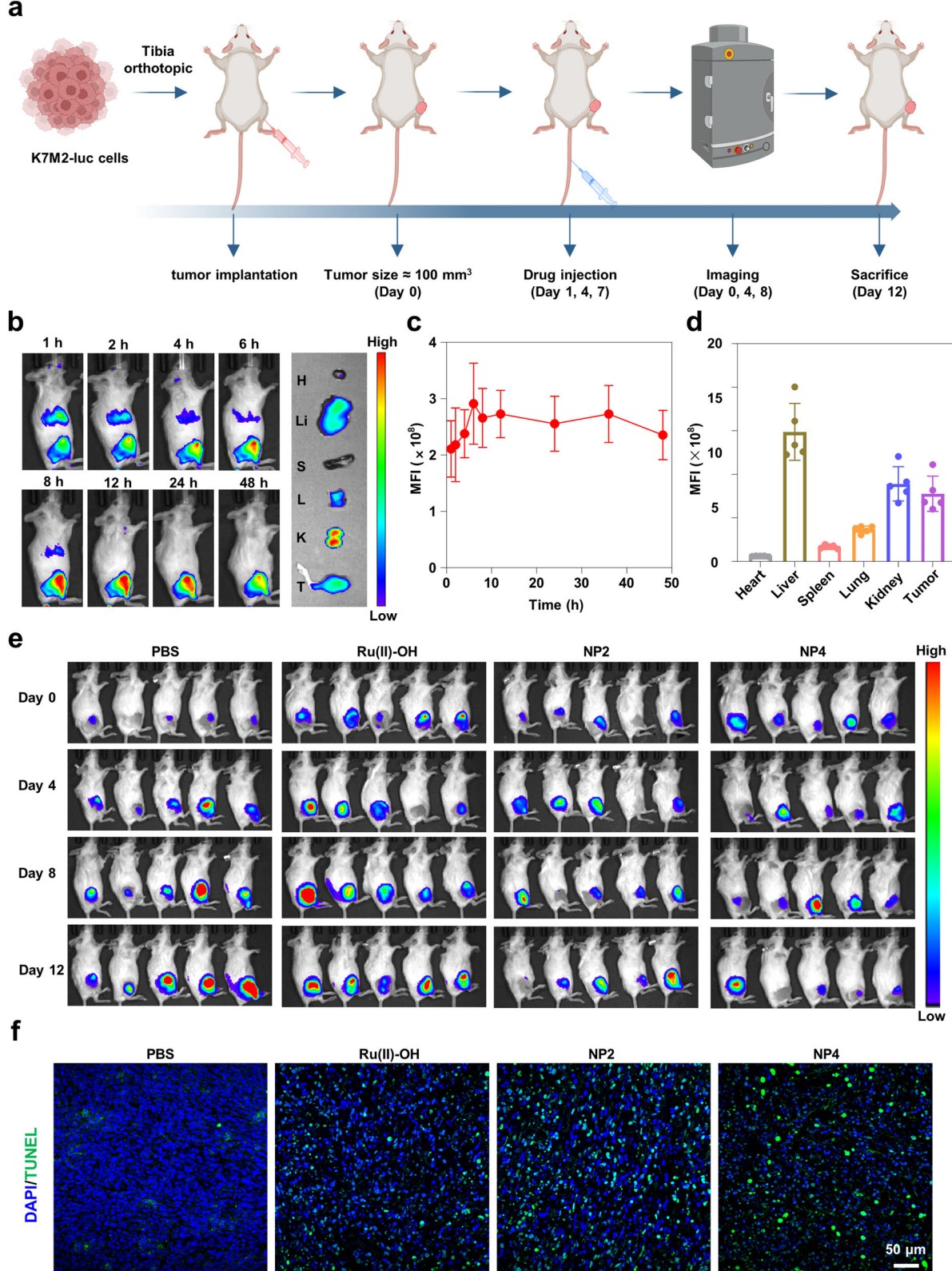

**Fig. 7 | The biosafety, biodistribution, and anti-tumor effect of NP4 in orthotopic K7M2 osteosarcoma model. a** Schematic treatment schedule. **b** Time-dependent imaging of the animal model upon injection of NP4-Cy7.5 (H heart, Li liver, S spleen, L lung, K kidney, T tumor). **c** Quantification from (**b**) (n = 5 mice). **d** Biodistribution inside the animal model 48 h after the injection of NP4-Cy7.5 (n = 5 mice). **e** Bioluminescence imaging of orthotopic K7M2 osteosarcoma model after receiving PBS, Ru(II)-OH, NP2, and NP4 in vivo (n = 5 mice). **f** Tumor tissue slices were stained with Terminal deoxynucleotidyl transferase dUTP nick end labeling (TUNEL) (green) and DAPI (blue) following different treatments. Figure 7a Created with BioRender.com released under a Creative Commons Attribution-NonCommercial-NoDerivs 4.0 International license.

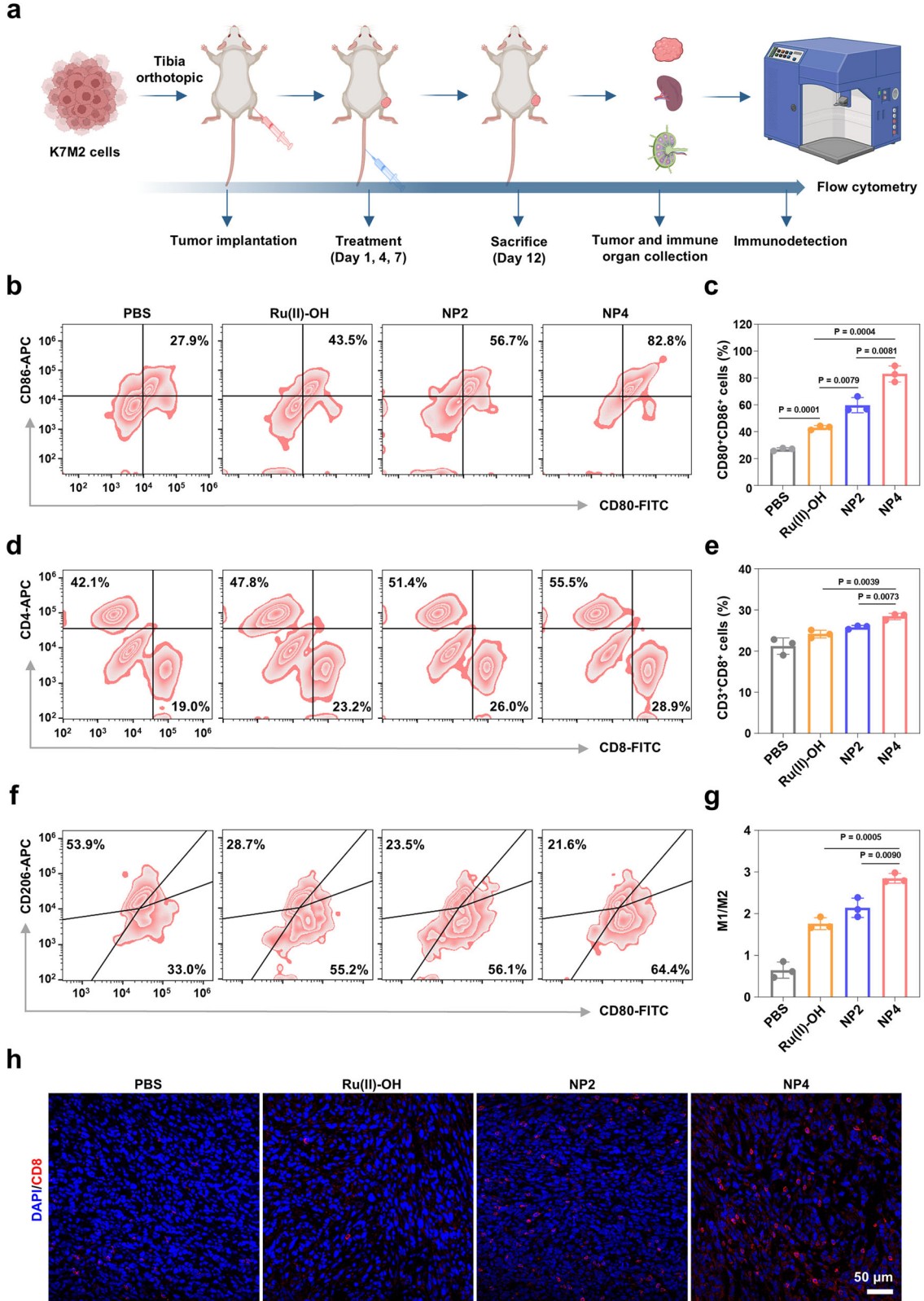

**Fig. 8 | The activation of immune responses by NP4 in vivo. a** Schematic treatment schedule. **b, c** The ratio of matured dendritic cells (CD80⁺, CD86⁺) in TDLNs collected from K7M2 mouse models subjected to different treatments. (n = 3 mice). **d, e** The ratio of effector T cells (CD8⁺ T cells and CD4⁺ T cells) collected from K7M2 mouse models subjected to different treatments. (n = 3 mice). **f, g** The ratio of M1-type macrophage (F4/80, CD80⁺) and M2-type macrophage (F4/80, CD206⁺) in tumor tissue collected from K7M2 mouse models subjected to different

treatments. (n = 3 mice). **h** Immunofluorescence imaging of the infiltration of CD8⁺ T cells in tumor tissues after various treatments. Data are presented as mean ± SD. n = 3 mice. Statistical significance between every two groups was calculated by T-test, the statistical test used was two-sided. Figure 8a Created with BioRender.com released under a Creative Commons Attribution-NonCommercial-NoDerivs 4.0 International license.

(Fig. 9a). The biodistribution of NP4-Cy7.5 was investigated using IVIS. The fluorescence intensity at the tumor site increased with a prolongation of the incubation time. The maximal emission was reached 36 h after injection of the nanoparticles (animal imaging: Fig. 9b, quantification at the tumor: Fig. 9c). At 60 h-post injection, the mice were sacrificed and the biodistribution determined upon fluorescence imaging of the respective organs. The results demonstrated that NP4-Cy7.5 primarily accumulated in liver, kidney, and the tumor (Fig. 9d). The tumor growth inhibition properties of Ru(II)-OH, NP2, or NP4 (2 mg Ru/kg) were studied upon intravenous injection in the tail vein of 143B tumor bearing mouse models. The tumor volume was recorded every 3 days. The results showed that the treatment with Ru(II)-OH reduced the tumor volume by 35%, the treatment with NP2 by 66%, and the treatment with NP4 by 91% (Fig. 9e). These results suggest the high tumor growth inhibition effect of NP4 against aggressive osteosarcoma inside the animal model. The mice models behave normally without signs for pain, stress, or discomfort and did not lose any weight (Fig. 9f). After the treatment, the mice were sacrificed, the tumor tissues were obtained and analyzed (photographs of the tumor: Supplementary Fig. 41, tumor weight: Fig. 9g). The tumor of the animals treated with NP4 showed a strong tumor growth inhibition effect. Subsequently, the tumor tissue was pathologically analyzed. The microscopy images showed cell rupture and lysis (H&E stain), indicative of signs of cell death (TUNEL stain) in the tumor tissue (Fig. 9h). Overall, these results suggest the high potential of NP4 as tumor-targeting anticancer agents.

## Therapeutic effects in patient-derived osteosarcoma xenograft mouse model

For a better understanding of the clinical potential, its therapeutic properties were assessed in a patient-derived osteosarcoma xenograft mouse model (PDX[OS]) in immunodeficient nude mice (protocol for model establishment: Fig. 10a). While the tumors treated with PBS showed an exponential growth, the tumor of the animals treated with Ru(II)-OH, NP2, or NP4 showed a tumor growth inhibition effect. Importantly, the treatment with NP4 showed the strongest therapeutic effect (individual tumor growth inhibition curves: Fig. 10b, combined tumor growth inhibition curves: Fig. 10c). Importantly, the animal models treated with the nanoparticles did not show any signs for pain, stress, or discomfort, and did not lose any weight (Fig. 10d), suggestive of the high biocompatibility of the treatment. Fifteen days after the treatment, all the mice were sacrificed and the tumorous tissue obtained (photographs of the tumor: Fig. 10e, tumor weight: Fig. 10f), confirming the strong therapeutic effect of NP4. H&E of the tumorous tissue of the animal model treated with NP4 showed high amounts of cell death and nuclear fragmentation (Fig. 10g, top). TUNEL staining further demonstrated high number of apoptotic cells inside the tumorous tissue of the animal model treated with NP4 (Fig. 10g, bottom). For an additional insight into the mechanism of action, the tumorous tissue was stained with a ROS-specific dye and analyzed by CLSM. While the animal model treated with PBS did not show any signs for ROS, high amounts of fluorescence signals in the tissue from the animal treated with NP4 were observed (Supplementary Fig. 42). Combined these findings demonstrated the high potential of the NP4 to eradicate challenging patient-derived osteosarcoma tumors inside an animal model.

## Discussion

In summary, this study reports on the rational design, synthesis, and in-depth biological evaluation of tumor-targeting ruthenium containing nanoreactors for glutathione oxidation catalysis in osteosarcoma cells with a 1% $O_2$ atmosphere. Using density functional theory calculations, the catalytic cycle of the ruthenium catalyst was predicted. Experimental evaluations demonstrated that the metal complex could efficiently catalyze the oxidation of glutathione (GSH)

into glutathione disulfide (GSSG). To enhance the pharmacological properties of the ruthenium catalyst, provide cancer selectivity for the treatment, and enable the treatment of challenging hypoxic tumors, the metal complexes were covalently conjugated to the side chains of polymers with a carbon fluorinated moiety that could co-deliver oxygen to the tumor. Based on the amphiphilic nature, the polymeric material could self-assemble into tumor-targeting nanoreactors. While the molecular ruthenium catalyst and the nanoreactors without carbon fluorinated moieties were cytotoxic in the low micromolar range in a 21% $O_2$ atmosphere, these compounds lost their therapeutic effect in a 1% $O_2$ atmosphere. Contrary, the ruthenium catalyst and carbon fluorinated moiety containing nanoreactors induced cytotoxic effect in a 1% and 21% $O_2$ atmosphere. In-depth study of mechanisms of action revealed that the nanoreactors could catalyze the oxidation of GSH into GSSG. The miss regulation of the GSH levels caused a disruption of the redox homeostasis and the antioxidation defense system, yielding the accumulation of ROS and lipid peroxides. The released ROS activate the ion channel TRPM2 in macrophages, promoting calcium influx, thereby eliciting an immune response. Based on these promising research results, the therapeutic effect of the nanoreactors were studied in osteosarcoma bearing mouse models. Upon intravenous injection into the tail vein, the nanoparticles selectively accumulated in the tumor of the animal model. The treatment with the nanoreactors demonstrated to nearly fully eradicate an aggressive osteosarcoma inside the animal model. Concurrently, the nanoreactors also activate the systemic immune system in mice, preventing tumor metastasis and recurrence. We are confident that here reported nanoparticles and structural derivatives could be suitable for clinical studies. The strategy of covalently conjugating metal-based anticancer agents to amphiphilic polymers, that could self-assemble into nanoparticles, could present a method for the tumor-selective anticancer therapy. The use of catalytic anticancer agents could reduce the therapeutic doses needed to achieve the desired therapeutic effect. The incorporation of carbon fluorinated moieties into the polymer backbone could allow for the codelivery of oxygen and therefore enable the treatment of hypoxic tumors.

## Methods

### Ethical approval

All animal experiments reported herein were performed following the guidelines and were approved by Peking University Institutional Animal Care and Use Committee (LA2021316). Patient-derived xenograft models were conducted in accordance with established guidelines, having obtained written consent from the respective patients, and received approval from the Second Xiangya Hospital of Central South University (2020591). The mice were euthanized through CO2 inhalation when the tumor volume surpassed 1500 mm³ or when the mice showed signs of serious weight loss, dreadful weakness, or Unhealed ulcers, indicating they were moribund. During treatment of the PDX model of osteosarcoma, since the individuality of the mice, the tumor volume of one mouse reached 1500 mm³ by the end of the treatment, and this mouse was not euthanized in time to obtain reliable and intuitive experimental results. However, the status of the mice was closely monitored during the subsequent treatment and no significant weight loss or extreme weakness was detected. In addition, the veterinarian was notified of the situation, and approval was obtained from the Ethics Committee.

### Animal experiment

BALB/c mice (4–5 weeks old, female) and BALB/c nude mice (4–5 weeks old, female) were purchased from SPF Biotechnology Co., Ltd. (Beijing, China). All mice were housed under a 12 h light/dark cycle, with the temperature consistently maintained between 18 °C and 23 °C and the humidity kept at ~50%.

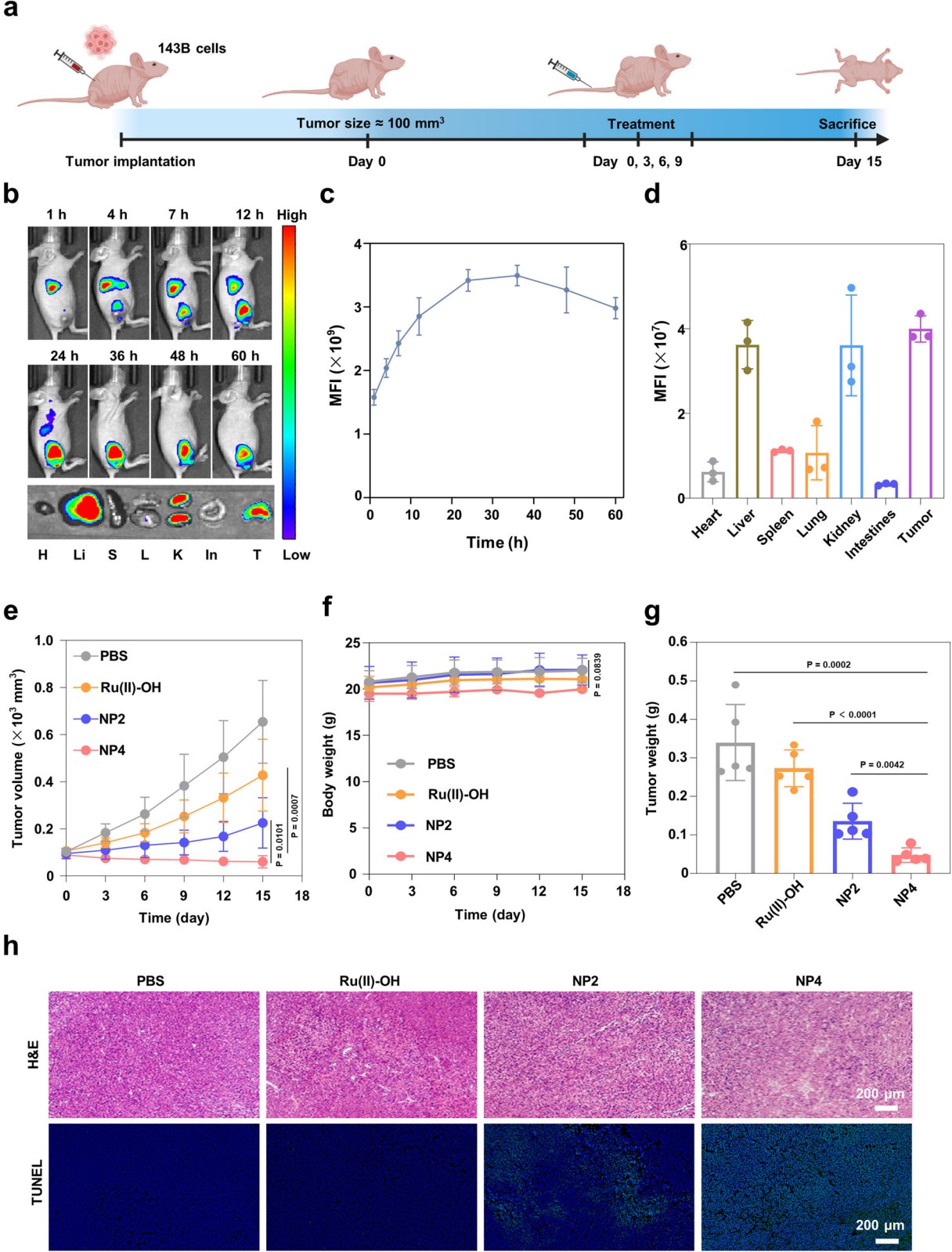

**Fig. 9 | Evaluation of the therapeutic properties of Ru(II)-OH, NP2, or NP4 (2 mg Ru/kg) upon intravenous injection in the tail vein of 143B tumor bearing mouse model. a** Schematic treatment schedule. **b)** Time-dependent imaging of the animal model upon injection of NP4-Cy7.5 (H heart, Li liver, S spleen, L lung, K kidney, In intestine, T tumor). **c** Quantification from (**b**) (n = 3 mice). **d** Biodistribution inside the animal model 60 h after the injection of NP4-Cy7.5 (n = 3 mice). **e** Tumor growth inhibition curves upon treatment (n = 5 mice). **f** Body weight upon treatment (n = 5 mice). **g** Tumor weight upon treatment (n = 5 mice). **h** Hematoxylin and eosin (H&E) and TUNEL staining of tumor tissues. Data are presented as mean ± SD. Statistical significance between every two groups was calculated by T-test, the statistical test used was two-sided. Figure 9a Created with BioRender.com released under a Creative Commons Attribution-NonCommercial-NoDerivs 4.0 International license.

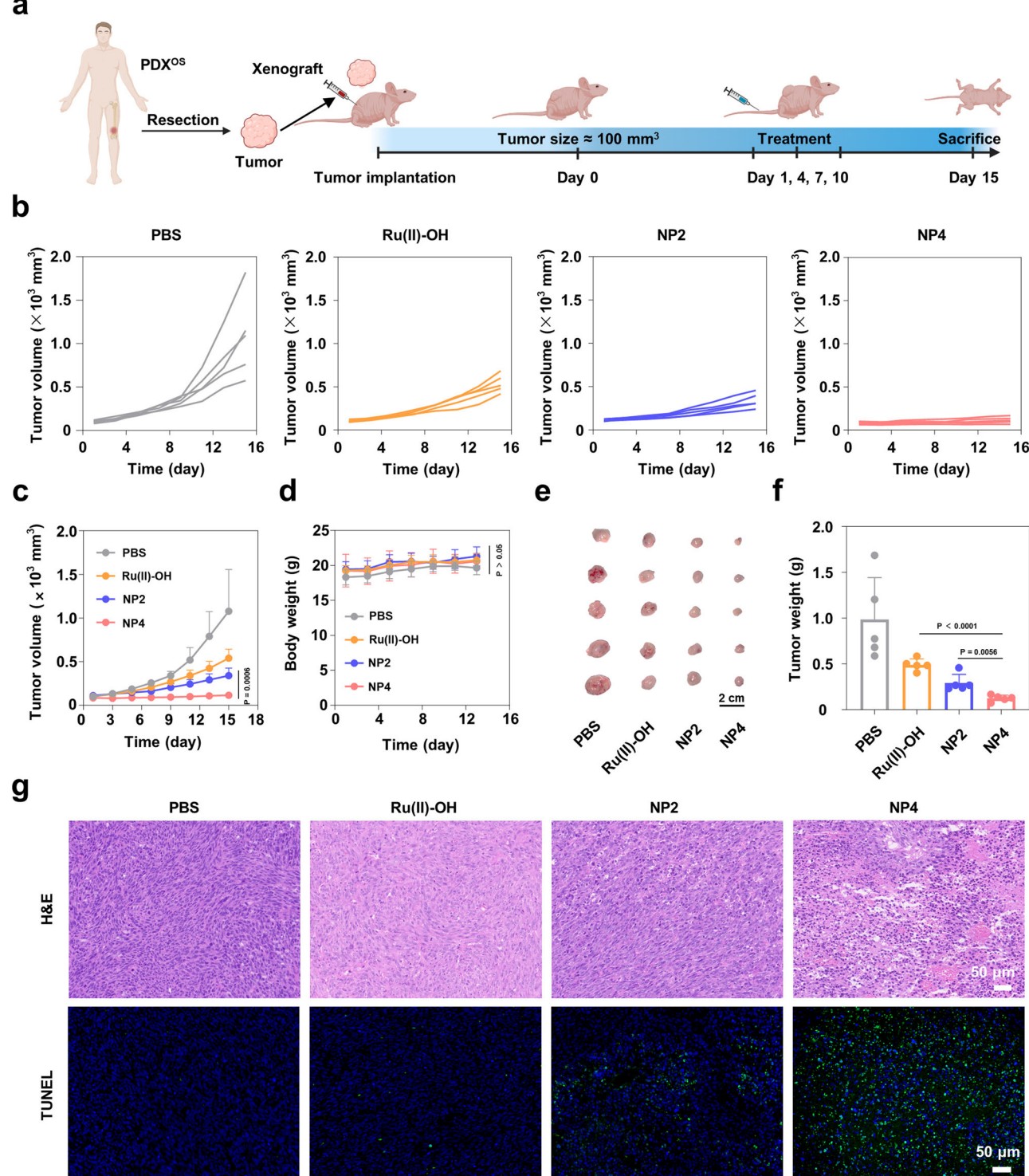

**Fig. 10 | Evaluation of the therapeutic properties of Ru(II)-OH, NP2, or NP4 (2 mg Ru/kg) upon intravenous injection in the PDX^OS model. a** Schematic illustration of the establishment of PDX model as well as the subsequent treatment schedule. **b**, **c** Tumor growth inhibition curves upon treatment. n = 5 mice. **d** Body weight upon treatment. n = 5 mice. **e** Photograph of tumors isolated from mice after different treatments. **f** Average tumor weight of the mice 15 days after treatment.

n = 5 mice. **g** H&E and TUNEL stain of the tumor tissues. Data are presented as mean ± standard deviation. n = 5 mice. Data are presented as mean ± SD. Statistical significance between every two groups was calculated by T-test, the statistical test used was two-sided. Figure 10a Created with BioRender.com released under a Creative Commons Attribution-NonCommercial-NoDerivs 4.0 International license.

## Sex of the animal models

To the best of our knowledge, the occurrence of osteosarcoma is not related to the sex of the patient. Therefore, in this work, the biological properties of the compounds were studied only in female mice.

## Material

Potassium iodide, 2-hydrazinylpyridine, hydroquinone, 3-bromo-1-propanol, ruthenium (III) chloride hydrate, γ-terpinene, ammonium hexafluorophosphate, 2,2,3,3,4,4,5,5,6,6,7,7,8,8,9,9-hexadecafluorodecane-1,10-diol, 1,2,4,5-cyclohexanetetracarboxylic dianhydride, mPEG5000-OH,

pyridine hydrofluoride, acetone, and mercaptoacetic acid were obtained from Aladdin (Shanghai, P. R. China). 3-(4,5-Dimethylthiazol-2-yl)-2,5-diphenyltetrazolium bromide (MTT) was obtained from Energy Chemical (Shanghai, China). Fetal bovine serum (FBS) was obtained from MeilunBio. RPMI-1640 medium, high-glucose DMEM were obtained from Thermo Fisher Scientific Inc. trypsin-EDTA (0.25%) and L-Glutamine-Penicillin-Streptomycin solution were obtained from Wuhan Servicebio Technology Co., Ltd.

## Instrumentation

$^1$H NMR spectra were measured by a 400 MHz NMR spectrometer (Bruker) at room temperature. Inductively coupled plasma mass spectrometer (Agilent technologies 7700 series, USA) was used for quantitative analysis of the total ruthenium contents from different samples and cells. The morphology and size of nanoparticles were obtained by transmission electron microscopy (TEM) carried out with a HT7700 and JEM-F200 electron microscopy. Localization NPs were performed using a confocal laser scanning microscopy (LSM-880, ZEISS). Size and zeta potential measurements were conducted on a Malvern Zetasizer (Nano ZS, UK). High resolution mass spectrometry (HR-MS) was conducted by Agilent 1290 UPLC/6540 Q-TOF. MTT assay was conducted using a microplate reader (SpectraMax). Ultra performance liquid chromatography (UPLC) analysis was performed on an Agilent 1290 series instrument. Graphite furnace atomic absorption spectrometry (GF-AAS, PerkinElmer PinAAcle D900 series, USA) was used for quantitative analysis of the total ruthenium contents from different samples and cells.

## Theoretical calculations

The geometry of a metal complex was determined using density-functional theory calculations with the Gaussian software package. (Frisch, M. J.; Trucks, G. W.; Schlegel, H. B.; Scuseria, G. E.; Robb, M. A.; Cheeseman, J. R.; Scalmani, G.; Barone, V.; Petersson, G. A.; Nakatsuji, H.; Li, X.; Caricato, M.; Marenich, A. V.; Bloino, J.; Janesko, B. G.; Gomperts, R.; Mennucci, B.; Hratchian, H. P.; Ortiz, J. V.; Izmaylov, A. F.; Sonnenberg, J. L.; Williams; Ding, F.; Lipparini, F.; Egidi, F.; Goings, J.; Peng, B.; Petrone, A.; Henderson, T.; Ranasinghe, D.; Zakrzewski, V. G.; Gao, J.; Rega, N.; Zheng, G.; Liang, W.; Hada, M.; Ehara, M.; Toyota, K.; Fukuda, R.; Hasegawa, J.; Ishida, M.; Nakajima, T.; Honda, Y.; Kitao, O.; Nakai, H.; Vreven, T.; Throssell, K.; Montgomery Jr., J. A.; Peralta, J. E.; Ogliaro, F.; Bearpark, M. J.; Heyd, J. J.; Brothers, E. N.; Kudin, K. N.; Staroverov, V. N.; Keith, T. A.; Kobayashi, R.; Normand, J.; Raghavachari, K.; Rendell, A. P.; Burant, J. C.; Iyengar, S. S.; Tomasi, J.; Cossi, M.; Millam, J. M.; Klene, M.; Adamo, C.; Cammi, R.; Ochterski, J. W.; Martin, R. L.; Morokuma, K.; Farkas, O.; Foresman, J. B.; Fox, D. J. Gaussian 16 Rev. C.01, Wallingford, CT, 2016) The metal atom was described using the Los Alamos (LANL2) effective core potential with the corresponding triple-zeta basis set while all other atoms were described with the Pople double-zeta basis set with a single set of polarization functions on non-hydrogen atoms (6-31 G(d)). The geometry of the calculated structures was verified by comparison with crystal structures of structurally related compounds from the CCDC. Solvent effects were included using a polarizable continuum model (PCM). The structures of all calculated molecules correspond to ground state minima on the ground state potential energy surfaces with no imaginary frequencies present.

## Synthesis of 4-(pyridin-2-yldiazenyl)phenol (compound 1)

The 4-(pyridin-2-yldiazenyl)phenol was synthetized using a similar procedure as previously reported[45]. 1,4-benzenediol (493 mg, 4.56 mmol) was dissolved in a mixture of water (50 mL) and 60% perchloric acid (3.6 mL). A solution of 2-hydrazine pyridine (504 mg, 4.62 mmol) in water (8 mL) was dropwise added. The

solution was stirred at room temperature for 1 h. Subsequently, an orange precipitate was formed that collected by filtration. The solid was washed with water (30 mL) and diethyl ether (30 mL). The compound was dried under high vacuum. Yield: 800 mg, 88%. $^1$H NMR (400 MHz, DMSO-d6) δ = 10.49 (s, 1H), 8.67 (dd, $J$ = 5.2, 1.8 Hz, 1H), 8.00 (td, $J$ = 7.7, 1.9 Hz, 1H), 7.92 – 7.83 (m, 2H), 7.66 (d, $J$ = 8.1 Hz, 1H), 7.51 (ddd, $J$ = 7.3, 4.8, 1.1 Hz, 1H), 7.06 – 6.89 (m, 2H). $^{13}$C NMR (100 MHz, DMSO-d$_6$) δ = 163.5, 162.4, 149.6, 149.6, 145.7, 139.3, 139.2, 126.0, 125.4, 116.6, 113.4. The analytical data was found to be in agreement with the previous literature[45].

## Synthesis of 3-(4-(pyridin-2-yldiazenyl)phenoxy)propan-1-ol (compound 2)

The 3-(4-(pyridin-2-yldiazenyl)phenoxy)propan-1-ol was synthetized using a similar procedure as previously reported[46]. 4-(pyridin-2-yldiazenyl)phenol (200 mg, 1 mmol) was dissolved in a mixture of DMF (10 mL), potassium carbonate (207 mg, 1.5 mmol), and potassium iodide(183 mg, 1.1 mmol). The bromopropanol (207 mg, 1.5 mmol) was added to the solution. The solution was stirred at 80°C for 24 h. After this time, the color of solution became blood red, and distilled water (200 ml) was added. The red solution was extracted three times with dichloromethane. The organic solvent was collected and concentrated. The products were obtained by silica gel column chromatography. $^1$H NMR (400 MHz, Chloroform-d) δ = 8.65 (dd, $J$ = 4.8, 2.6 Hz, 1H), 8.04–7.93 (m, 2H), 7.83 (td, $J$ = 7.7, 2.2 Hz, 1H), 7.74 (dd, $J$ = 8.1, 2.6 Hz, 1H), 7.32 (p, $J$ = 3.3 Hz, 1H), 7.02–6.91 (m, 2H), 4.15 (dt, $J$ = 8.6, 3.9 Hz, 2H), 3.83 (dt, $J$ = 8.5, 3.7 Hz, 2H), 2.03 (dt, $J$ = 8.4, 3.9 Hz, 2H). $^{13}$C NMR (100 MHz, Chloroform-d) δ = 163.0, 162.5, 149.3, 146.8, 138.4, 125.7, 124.7, 115.0, 114.8, 65.6, 59.4, 32.0. The analytical data was found to be in agreement with the previous literature[46].

## Synthesis of dichloro(p-cymene)ruthenium(II) dimer (compound 3)

The dichloro(p-cymene)ruthenium(II) was synthetized using a similar procedure as previously reported[47]. The ruthenium chloride hydrate (128 mg, 0.57 mmol) was dissolved in ethanol anhydrous (6.2 mL). γ-terpinene (425 μL, 2.65 mmol) was added into solution. The obtained solution was stirred at 80 °C for 2 h. Subsequently, the solution was cooled to room temperature. An orange precipitate formed that collected by filtration. The products were dried under high vacuum. Yield: 800 mg, 90%. $^1$H NMR (400 MHz, DMSO-d$_6$) δ = 5.82 (d, $J$ = 6.1 Hz, 2H), 5.78 (d, $J$ = 6.2 Hz, 2H), 2.84 (hept, $J$ = 7.0 Hz, 1H), 2.09 (s, 3H), 1.20 (d, $J$ = 6.9 Hz, 6H). $^{13}$C NMR (100 MHz, DMSO-d6) δ = 106.9, 100.6, 86.8, 86.0, 30.5, 22.0, 21.9, 18.4, 18.3. The analytical data obtained were found to be consistent with the previous literature[47].

## Synthesis of diiodo(p-cymene)ruthenium(II) dimer (compound 4)

The diiodo(p-cymene)ruthenium(II) dimer was synthetized using a similar procedure as previously reported[12]. The dichloro(p-cymene)ruthenium(II) (1.06 mmol, 650 mg) was dissolved in water (250 mL). The solution was heated reflux for 1 h, filtered instantly. Potassium iodide (4.45 g, 26.8 mmol) was added to the filtrate under stirring conditions. The brown-red precipitate formed that collected by filtration. The solid was washed with ethanol (30 mL) and ether (30 mL). A purplish-red solid product was obtained. Yield: 941 mg, 91%. $^1$H NMR (400 MHz, DMSO-d$_6$) δ = 5.85 (q, $J$ = 6.3 Hz, 4H), 3.15 (hept, $J$ = 6.9 Hz, 1H), 2.39 (s, 3H), 1.21 (d, $J$ = 6.9 Hz, 6H). $^{13}$C NMR (100 MHz, DMSO-d$_6$) δ = 110.6, 102.5, 88.3, 86.5, 31.8, 22.5, 20.5, 20.4. The analytical data obtained were found to be consistent with the previous literature[12].

## Synthesis of compound Ru(II)-OH

The diiodo(p-cymene)ruthenium(II) dimer (54.8 mg, 0.05 mmol) was dissolved in methanol (20 mL) under 40 °C. A solution of 3-(4-(pyridin-2-yldiazenyl)phenoxy)propan-1-ol (26 mg, 0.1 mmol) in methanol (10 mL) was dropwise added. The reaction solution gradually becomes black, stirred at room temperature for 3 h, the solvent is concentrated to 10 mL by distillation under reduced pressure. Ammonium hexafluorophosphate (83 mg, 0.5 mmol) was added to the solution, which was then stirred for 1 h. The brown-black solid product was obtained through silica gel column chromatography. Yield: 44 mg, 57%. $^1$H NMR (400 MHz, Chloroform-d) δ = 9.30 (d, $J$ = 5.7 Hz, 1H), 8.50 (d, $J$ = 7.9 Hz, 1H), 8.21 – 8.07 (m, 3H), 7.69 (t, $J$ = 6.8 Hz, 1H), 7.09 (dd, $J$ = 9.3, 4.7 Hz, 2H), 6.17 (d, $J$ = 6.3 Hz, 1H), 5.85 (dd, $J$ = 15.4, 6.5 Hz, 2H), 5.69 (d, $J$ = 6.3 Hz, 1H), 4.37 – 4.28 (m, 2H), 3.92 (s, 2H), 2.50 (s, 3H), 2.33 – 2.21 (m, 1H), 2.14 (p, $J$ = 6.1 Hz, 2H), 1.08 (d, $J$ = 6.9 Hz, 3H), 0.99 (d, $J$ = 6.9 Hz, 3H). $^{13}$C NMR (100 MHz, DMSO-d$_6$) δ = 164.6, 163.5, 156.2, 151.5, 141.8, 128.5, 127.9, 127.7, 115.5, 111.7, 108.7, 91.6, 90.7, 89. 6, 66.3, 57.5, 22.2, 21.7, 20.9. HR-ESI (positive ion mode): 620.0349 m/z [M]$^+$; calcd for C$_{24}$H$_{29}$IN$_3$O$_2$Ru$^+$ 620.0343 m/z [M]$^+$.

## Synthesis of the 2,2'-(propane-2,2-diylbis(sulfanediyl)) bis(ethan-1-ol)

The 2,2'-(propane-2,2-diylbis(sulfanediyl))bis(ethan-1-ol) was synthetized using a similar procedure as previously reported[48]. Mercaptoacetic acid (19.82 g, 215.15 mmol) and trifluoroacetic acid (TFA, 100 μL) were dissolved in acetone (5.00 g, 86.09 mmol). The reaction was carried out at 25 °C for 24 h. Then, acetonitrile was added at 85 °C until the components were dissolved completely. The resulting acetonitrile solution was slowly cooled to give 2,2'-(propane-2,2-diylbis(sulfanediyl))diacetic acid.

Lithium aluminum hydride (7.50 g, 197.63 mmol) was added into a solution of 2,2'-(propane-2,2-diylbis(sulfanediyl))diacetic acid (11.10 g, 49.55 mmol) in dry tetrahydrofuran (200 mL). The mixture was stirred at room temperature for 12 h. Subsequently, the reaction system was diluted with ethyl acetate (200 mL × 3), followed by washing with water (100 mL × 3) and saturated sodium chloride solution (100 mL × 3). The organic layer was dried over anhydrous Na$_2$SO$_4$, filtered, concentrated in vacuo, and purified by silica gel column chromatography to give light-yellow oil pruducts 2,2'-(propane-2,2-diylbis(sulfanediyl)) bis(ethan-1-ol). Yield: 4.25 g, 43.8%. $^1$H NMR (400 MHz, DMSO-d6) δ = 4.78 (s, 2H), 3.52 (t, $J$ = 7.0 Hz, 4H), 2.66 (t, $J$ = 7.1 Hz, 4H), 1.53 (s, 6H). $^{13}$C NMR (100 MHz, DMSO-d$_6$) δ = 61.2, 55.8, 33.2, 31.4. The analytical data obtained were found to be consistent with the existing literature[48].

## Synthesis of ROS-sensitive polymers (P1)

To a solution of 1h,1h,10h,10h-perfluoro-1,10-decanediol (0.62 g, 1.35 mmol) and ROS sensitive molecular (2,2'-(propane-2,2-diylbis (sulfanediyl))bis(ehan-1-ol)) (0.26 g, 1.35 mmol) in anhydrous DMF (5 mL), 1,2,4,5-cyclohexanetetracarboxylic dianhydride (0.63 g, 2.83 mmol) was quickly added. Following 8 h of magnetic stirring at 25 °C, mPEG5000-OH (1.35 g, 0.27 mmol) was incorporated into the reaction system. The stirring continued for an additional 12 h at 50 °C, after which the mixture was introduced to 10 mL of deionized water while being sonicated. Subsequently, the solution was subjected to dialysis using a dialysis bag with a molecular weight cutoff of 8000–14,000 Da. After 72 h, the resulting solution was freeze-dried by using low-temperature freeze dryers, yielding P1(1.23 g) as a yellow powder. $^1$H NMR (400 MHz, DMSO-d$_6$) was shown in Supplementary Fig. 16.

## Synthesis of ROS-sensitive polymers conjugated with Ru(II)-OH (P2)

4-Dimethylaminopyridine (DMAP, 126 mg) was added at room temperature to a solution of P1 (300 mg) and 1-(3-Dimethylaminopropyl)-3-ethylcarbodiimide hydrochloride (EDCl, 200 mg). Following an additional 1 h of stirring at 25 °C, Ru(II)-OH (50 mg) was introduced into the reaction system. After an additional 12 h of stirring at room temperature, the solution was introduced into 10 mL of deionized water while undergoing sonication. This was followed by dialysis using a dialysis bag with a molecular weight cutoff of 8000-14000 Da. Following a 72 h dialysis period, the solution was freeze-dried by using low-temperature freeze dryers, resulting in P2 (287 mg), which appeared as a brown powder. $^1$H NMR (400 MHz, DMSO-$d_6$) was shown in Supplementary Fig. 18.

## Synthesis of ROS-sensitive polymers (P3)

To a solution of ROS sensitive compound (2,2'-(propane-2,2-diylbis (sulfanediyl))bis(ehan-1-ol)) (0.26 g, 1.35 mmol) in anhydrous DMF (5 ml), 1,2,4,5-cyclohexanetetracarboxylic dianhydride (0.31 g, 1.41 mmol) was quickly added into the solution. After undergoing magnetic stirring for 8 h at room temperature, mPEG$_{5000}$-OH (0.68 g, 0.13 mmol) was introduced into the reaction system. The stirring process continued for another 12 h at 50 °C, after which the mixture was introduced into 10 mL of deionized water while being sonicated. Following this, the solution was subjected to dialysis using a dialysis bag with a molecular weight cutoff of 8000-14000 Da. After 72 h of dialysis, the resulting solution was freeze-dried by using low-temperature freeze dryers, yielding P3 (0.62 g), which was obtained as a white powder. $^1$H NMR, $^{19}$F NMR (400 MHz, DMSO-d$_6$) was shown in Supplementary Fig. 19.

## Synthesis of ROS-sensitive polymers conjugated with Ru(II)-OH (P4)

At room temperature, 4-Dimethylaminopyridine (DMAP, 126 mg) was introduced to a solution of Ru(II)-OH (50 mg) and 1-(3-Dimethylaminopropyl)-3-ethylcarbodiimide hydrochloride (EDCl, 200 mg). After stirring for another 1 h at 25 °C, P3 (300 mg) was introduced to the reaction mixture. After an 1 h of magnetic stirring at room temperature, P3 (300 mg) was introduced into the reaction system. The stirring continued for another 12 h at room temperature, after which the mixture was transferred into 10 mL of deionized water while being sonicated. Subsequently, the solution underwent dialysis using a dialysis bag with a molecular weight cutoff of 8000–14,000 Da. Following 72 h of dialysis, the resulting solution was by using low-temperature freeze dryers, which yielded P4 (301 mg), presented as a brown powder. $^1$H NMR (400 MHz, DMSO-d$_6$) was shown in Supplementary Fig. 21.

## General procedure for experimental methods

**NMR analysis.** The conversion of GSH into GSSG was monitored by NMR (Bruker Avance 400). $^1$H-NMR spectra of GSH (50 mM) in the presence of Ru(II)-OH (2 mM) were directly conducted after preparation and after incubation for 12 h.

**Mass spectrometry.** The conversion of GSH into GSSG was monitored by mass spectrometry (Agilent 1290 UPLC/6540 Q-TOF). High resolution mass spectrometry of GSH (50 mM) in the presence of Ru(II)-OH (2 mM) were directly conducted after preparation and after incubation for 12 h. Each experiment was replicated twice.

**Ultra performance liquid chromatography (UPLC) analysis.** The conversion of GSH into GSSG was monitored by UPLC (Agilent 1290 Infinity II) equipped with a Poroshell 120 EC-C18 column (4.6 × 150 mm, 4 μm, 120 Å) at a flow rate of 1.0 mL/min at 37 °C. The UPLC mobile phase consisted of a linear gradient of the two mixtures of 0.1% TFA in water and 0.1% TFA in acetonitrile (ACN). The chromatogram was recorded via UV absorption at 210 nm.

**Catalytic activity of Ru(II)-OH.** For kinetic analysis, the redox reaction of glutathione (GSH) was conducted at 37 °C. Ru(II)-OH with a

concentration of 50 μM was incubated with different concentrations of GSH substrates (250 μM, 500 μM, 1000 μM and 2000 μM). The GSH and GSSG contents were analyzed after incubation for 0, 1, 2, 3, 4, 5, 6, 8, 10 and 12 h. The kinetic data were analyzed by using the GraphPad Prism software with a simple pseudo first-order reaction model to estimate the apparent kinetic parameters.

$$v = k1[S]$$

where "$S$" is the substrate (GSH), v is the reaction rate, k1 is the apparent first-order rate constant k1 was calculated based on the slope of fitted $v$-[$S$] curve.

To calculate the turnover number (TON) and the turnover frequency (TOF), 2000 μM GSH substrates were incubated with 50 μM Ru(II)-OH. The reaction mixtures were subjected to UPLC assay after 12 h incubation at 37 °C. The ratios of starting materials and the calculated TON and TOF in the reactions.

### Preparation of nanoparticles
P2 (100 mg) or P4 (100 mg) was dissolved in 1 mL DMSO, and the solution were then slowly added dropwise to 10 mL of water. Unloaded drugs and organic solvent were removed through dialysis against deionized water for 48 h. The obtained NP2 and NP4 were stored at 4 °C for subsequent use. The concentration of Ru(II)-OH in nanoparticles were assessed via ICP-MS.

Dyes (Cy5.5, Cy7.5, or Nile Red) and P4 (100 mg) were co-dissolved in 1 mL DMSO, and the solution were then slowly added dropwise to 10 mL of water. Unloaded dyes and organic solvent were removed through dialysis against deionized water for 48 h. The obtained NP4-Cy5.5, NP4-Cy7.5, and NP4-NR were stored at 4 °C for subsequent use.

### General characterization of nanoparticles
The size distribution of NP2 and NP4 was detected via a dynamic light scattering device (Malvern Zetasizer Nano, UK). The morphology and shape of NP2 and NP4 was measured with a TEM.

### Ru release of NP4
To examine the release profile of the nanoparticles, NP4 (Ru concentration: 800 μM, 3 mL) was placed in a dialysis bag with a molecular weight cutoff of 3500 Da. This dialysis bag was then immersed in either 200 mL of phosphate-buffered saline (PBS) or a 200 mL aqueous solution of hydrogen peroxide ($H_2O_2$) at a concentration of 10 mM, all while being maintained in a shaking culture incubator at 37 °C. At specified time intervals (0 h, 1 h, 2 h, 4 h, 6 h, 8 h, 12 h, 24 h, 48 h), the sample solution (1.5 mL) was withdrawn from the dialysate, after which an equal volume of fresh corresponding solution (1.5 mL) was promptly added back into the dialysate to maintain the overall volume. All collected samples were subsequently analyzed using ICP-MS. The ratio of ruthenium released from the nanoparticles was calculated as the percentage of cumulative ruthenium in the dialysate relative to the total amount of ruthenium present in the nanoparticles.

### Measurement of oxygen loading capacity of NP4
PBS, NP2, and NP4 were firstly bubbled with the flowing $N_2$ to exclude the content of $O_2$. Afterward, the PBS, NP2, and NP4 solutions were bubbled with the flowing $O_2$, where the $O_2$ concentration was monitored using a portable dissolved oxygen meter (Hanna S20 Instruments HI-2004-02 Edge® Dissolved Oxygen Meter) for 5 min. The values were recorded automatically every 30 s.

### Cell culture
All cell lines (K7M2 (CRL-2836), 143B (CRL-8303), HOS (CRL-1543)) were obtained from the American Type Culture Collection (ATCC). These cell lines are not on the list of known misidentified cell line maintained by the International Cell Line Authentication Committee. The 143B and HOS cell lines were cultured in RPMI 1640 medium supplemented with 10% FBS and 1% P/S. The K7M2 cells were maintained in DMEM, also supplemented with 10% FBS and 1% P/S. Mycoplasma tests were performed monthly and were found to be negative.

### Cell uptake of nanoparticles
The cellular uptake of nanoparticles was monitored by CLSM. Briefly, Poly-lysine cell crawls were placed in a 12-well plate. Following this, cell media containing 143B cells at a density of $3 × 10^5$ were added and incubated overnight in a cell culture incubator. Subsequently, the cells were incubated with Cy5.5-labeled NP4 for varying time intervals of 1 h, 4 h, or 7 h, respectively. After being washed with cold PBS, the cells were fixed with paraformaldehyde. Cell nuclei were stained with DAPI. Subsequently, images were collected with CLSM (LSM-800, ZEISS, Germany) (DAPI, $\lambda_{ex} = 405$ nm, $\lambda_{em} = 460$ nm, Cy5.5, $\lambda_{ex} = 673$ nm, $\lambda_{em} = 692$ nm).

Flow cytometry was further applied to observe the cellular uptake of nanoparticles. 143B cells were plated in 12-well plates at a density of $3 × 10^5$ cells/well and subsequently cultured for 12 h. Cell were then treated with Cy5.5-labeled NP4 for 1 h, 4 h, or 7 h, respectively, washed with PBS, and analyzed by flow cytometry (Beckman Coulter, U.S.A.).

### The uptake and apoptosis test in 3D tumor spheroids
To establish 3D tumor spheroids, 50 μL of a 1% agarose gel solution (Aladdin) was added into each well of a 96-well plate. Subsequently, 200 μL media containing 143B cells at a density of $2×10^3$ was added to each well. At day 7, the formation of cell spheres was completed. For analysis of cellular uptake of NP4, the 3D spheroids were treated with Cy5.5-labeled NP4 for 12 h. After washing with cold PBS, the cellular uptake of NP4 was measured via CLSM. For the apoptosis test, the spheroids were treated with PBS, Ru(II)-OH, NP2, or NP4 for 24 h, respectively (Ru 10 μM). After being washed with cold PBS, the spheroids were stained with Calcein AM/PI Cell Viability Kit (Beyotime). Subsequently, images were collected with CLSM.

### Cell viability assays
To evaluate cell viability, 143B and K7M2 cells were seeded in 96-well plates at a density of $5 × 10^3$ cells per well and cultured for 24 h. Subsequently, the cells were treated with PBS, Ru(II)-OH, NP2, or NP4 at various final concentrations ranging from 0.025 μM to 15 μM for 48 h. Subsequently, cells were then incubated with 10% MTT for 4 h before measurement at 570 nm.

### Apoptosis analysis
Cellular apoptosis was assessed with an Annexin V-FITC apoptosis detection kit (Beyotime) according to the manufacturer's instructions. In brief, 143B cells were seeded on 12-well plates at $3 × 10^5$ cells per well. After 12 h incubation, cells were treated with PBS, Ru(II)-OH, NP2, or NP4 for 24 h (Ru 10 μM), washed with PBS and incubated with Annexin/PI reagent in the dark for 15 min at 25 °C. Thereafter, the cells were immediately detected by flow cytometer.

### Intracellular hypoxia detection
Hypoxyprobe™-1 (Hypoxyprobe Inc, Burlington, USA) is a substituted 2-nitroimidazole compound known as pimonidazole. Since it can bind to cells if the $pO_2$ levels are less than 10 mm Hg, Hypoxyprobe™-1 is used as a probe for intracellular hypoxia detection. The 143B cells were cultured to 80% confluence, and incubated with 200 μM hypoxyprobe and PBS, Ru(II)-OH, NP2, or NP4 (Ru 10 μM) under hypoxia condition for 12 h. Following incubation, the 143B cells were fixed and blocked with 3% bovine serum albumin (BSA) and then treated with anti-hypoxyprobe antibody and DAPI. A blank control group was subjected to the same protocol. All treated 143B cells were subsequently analyzed using CLSM.

## Measurement of ROS production

To measure intracellular ROS, the 143B cells were plated in 12-well plates at a density of $3 \times 10^5$ cells per well. Following a 12 h incubation period, the 143B cells were incubated with PBS, Ru(II)-OH, NP2, or NP4 (10 μM Ru) for 6 h. The cells were subsequently treated with DCFH-DA (10 μM, Bio-lab) for 1 h in the dark, and washed three times with PBS. The fluorescence was then analyzed using flow cytometer and CLSM.

## Measurement of GSH/GSSG level

For GSH/GSSG level, 143B cells were plated in 12-well plates. Subsequently, the 143B cells were incubated with PBS, Ru(II)-OH, NP2, and NP4 (Ru 10 μM) for 6 h. The GSH and GSSG amount was evaluated using GSH/GSSG assay kit.

## Intracellular lipid peroxide (LPO) detection

To detect the intracellular LPO content, 143B cells were incubated with PBS, Ru(II)-OH, NP2, and NP4 (Ru 10 μM) for 6 h. Subsequently, the cells were washed with PBS and stained by C11-BODIPY$^{581/591}$ (5 μM) for 20 min. Then the cells were analyzed by CLSM and flow cytometer.

## Intracellular hydrogen peroxide (H2O2) detection

To detect the intracellular $H_2O_2$ content, 143B cells were incubated with PBS, Ru(II)-OH, NP2, and NP4 (Ru 10 μM) for 6 h. Subsequently, the cells were washed with PBS and stained by ROS Green$^{TM}$ $H_2O_2$ Probe. Then the cells were analyzed by CLSM.

## Measurement of intracellular and cell supernatant H$_2$O$_2$ levels

For intracellular and cell supernatant $H_2O_2$ levels, 143B cells were plated in 6-well plates. Subsequently, the 143B cells were incubated with PBS, Ru(II)-OH, NP2, and NP4 (Ru 10 μM) for 6 h. The intracellular and cell supernatant $H_2O_2$ amount was evaluated using $H_2O_2$ assay kit.

## Analysis of calcium ion (Ca$^{2+}$) levels in macrophages

To detect the intracellular $Ca^{2+}$ content, 143B cells were incubated with PBS, Ru(II)-OH, NP2, and NP4 (Ru 10 μM) for 6 h. The medium was replaced, and after 8 h the cell supernatant was collected and co-incubated with RAW264.7 cells for 4 h. Subsequently, the RAW264.7 cells were washed with PBS and stained by $Ca^{2+}$ Probe Fluo-4 AM for 20 min. Then the cells were analyzed by CLSM.

## Western blot assay

K7M2 cells were incubated with the PBS, Ru(II)-OH, NP2, or NP4 (10 μM Ru) for 12 h at 37 °C, and the supernatant was subsequently co-incubated with RAW264.7 cells. Then, the RAW264.7 cells were lysed using RIPA lysis buffer supplemented with 1 mM phenylmethanesulfonyl fluoride (PMSF). The cells were lysed by ultrasound for 20 min at 4 °C, followed by centrifugation at 12,000 rpm for 15 min to collect the supernatant. The protein concentration was quantified using a BCA kit, and then SDS-PAGE buffer was added. Following this, the protein lysates were denatured in a metal bath at 95 °C and subjected to SDS-polyacrylamide gel electrophoresis (SDS-PAGE) for separation. The separated proteins were then transferred onto a PVDF membrane (Merck Millipore) and blocked using skim milk for 2 h at room temperature. After blocking, the membrane was incubated overnight at 4 °C with the specified primary antibodies, including TRPM2 antibody at a dilution of 1:1000 and GAPDH antibody at a dilution of 1:3000. The PVDF membrane was subsequently washed (5 min × 5) with tris-buffered saline with tween (TBST) buffer and then incubated with the appropriate secondary antibody (Peroxidase-conjugated goat anti-rabbit IgG (H + L), at a dilution of 1:5000 in a 5% bovine serum albumin solution for 1 h at 25 °C. After further washing with TBST buffer, the membrane was photographed using the chemiluminescent imaging system.

## Biosafety assay of NP4

The biosafety of Ru(II)-OH and NP4 (Ru 800 μM, 200 μL) were evaluated in healthy female BABL/c mice. Twelve mice were randomly divided into 4 groups and injected into the tail vein with PBS, Ru(II)-OH, NP2, and NP4 (2 mg Ru/kg) on the days 1, 4, and 7. On day 14, the mice were euthanized, and a histological analysis of the major organs was performed using H&E staining.

## Pharmacokinetic assay of NP4

The pharmacokinetics of Ru(II)-OH and NP4 (Ru 800 μM, 200 μL) were evaluated in healthy female BABL/c mice (5 weeks). Six mice were randomly divided into 2 groups and injected into the tail vein with Ru(II)-OH and NP4, respectively, and 10 μL of blood was collected from the tail tip at 5 min, 10 min, 20 min, 30 min, 1 h, 2 h, 4 h, 8 h, 12 h, 24 h, and 48 h. Finally, the blood samples were nitrated with nitric acid, and the ruthenium content was determined by ICP-MS.

## Establishment of K7M2-Luc orthotopic osteosarcoma mouse model and therapeutic effect evaluation

**Tumor model establishment.** To establish orthotopic model, K7M2 cells ($3 \times 10^6$ cells/well) were dispersed in PBS buffer and implanted into right tibia of BALB/c mice.

**Biodistribution study.** The biodistribution of NP4 in orthotopic osteosarcoma mouse model was investigated using in vivo imaging system (IVIS). Ten 5-week-old female BALB/c mice bearing tumors were randomly assigned into two groups. When the tumor volume reached approximately 200 mm³, the mice were intravenously administered with Cy7.5-labeled NP4 at a dosage of 2 mg/kg. Following the injection, in vivo imaging was conducted using an IVIS Spectrum (PerkinElmer) at various time points: 1 h, 2 h, 4 h, 6 h, 8 h, 12 h, 24 h, and 48 h post-injection, utilizing an excitation wavelength of 780 nm and a fluorescence emission wavelength of 840 nm. At 48 h post-injection, the mice were euthanized to collect tumors and major organs for subsequent ex vivo imaging.

The biodistribution of NP4 was studied by using ICP-MS. Six 5-week-old female BALB/c mice bearing tumors were randomly assigned into two groups. When the tumor size reached approximately 200 mm³, the mice received intravenous injections of Ru(II)-OH and NP4 at a dosage of 2 mg Ru/kg. After 48 h, the mice were sacrificed, and the heart, liver, spleen, lung, kidney, and tumor tissues were subsequently isolated, weighed, and nitrated for further analysis. Finally, the ruthenium content in each organ and tumor tissue was quantified by ICP-MS.

**Therapeutic effect evaluation.** Twenty tumor-bearing female BALB/c mice, each aged 5 weeks, were randomly assigned into four groups, with each group comprising five mice. When the size of tumor reached 100 mm³, the mice were intravenous injected with PBS, Ru(II)-OH, NP2, and NP4 at the dose of 2 mg Ru/kg. The tumor volume was monitored through imaging using an IVIS Spectrum system (PerkinElmer) at 0-day, 4-day, 8-day, and 12-day postinjection, respectively (100 μL luciferin−per mouse, 15 mg ml⁻¹).

**Flow cytometry analysis of the animal tissue.** Female BALB/c mice (5 weeks old) were implanted with K7M2 cells ($3 \times 10^6$) in the right tibia. Once the tumor volume reached approximately 100 mm³, twelve mice were assigned at random (n = 3) and received the same treatment as previously described. Mice were sacrificed at day 12. The harvested tumors and draining lymph nodes were utilized to prepare single-cell suspensions. The cells from tumor tissue were stained with anti-CD3-PE, anti-CD4-APC, and anti-CD8-FITC antibodies for 30 min, and further analyzed the ratio of different types T cells by flow cytometry. the cells from tumor and draining lymph nodes tissue were stained with anti-CD11c-PE, anti-CD80-FITC, and anti-CD86-APC antibodies for

30 min, and further analyzed the ratio of mature dendritic cells (DCs). All antibodies used in this study were purchased from Dakewe Biotech Co., Ltd. Data acquisition and processing for flow cytometry were conducted using CytExpert software.

### Establishment of 143B subcutaneous osteosarcoma mouse model and therapeutic effect evaluation

**Tumor model establishment and biodistribution study of NP4 in vivo.** Three female BALB/c nude mice were subcutaneously inoculated with 143B human osteosarcoma cells ($3 \times 10^6$) on the right buttock. The tumor-bearing mice were intravenously administered Cy7.5-labeled NP4 at a dosage of 2 mg/kg (Ru) once the tumor volume reached approximately 200 mm³. Following the injection, imaging was performed using an IVIS Spectrum system (PerkinElmer) at various time points: 1 h, 4 h, 7 h, 12 h, 24 h, 36 h, 48 h, and 60 h post-injection, with an excitation wavelength of 780 nm and a fluorescence emission wavelength of 840 nm. At 60 h post-injection, the mice were euthanized in order to collect tumor tissues and major organs for subsequent ex vivo imaging.

### In vivo antitumor efficacy assessment

When the tumor volumes reached approximately 100 mm³, twenty female BALB/c nude mice bearing 143B tumors were randomly assigned into four groups. Subsequently, they received intravenous injections of PBS, Ru(II)-OH, NP2, or NP4 (at a dosage of 2 mg/kg Ru) on days 0, 3, 6, and 9, respectively. The tumor volume and body weight of all the mice were recorded every two days. The tumor volume (in mm³) was calculated using the formula V = (a × b²)/2, where "a" represents the length and "b" denotes the width of the tumor.

### Establishment of PDXOS model and therapeutic effect evaluation

The osteosarcoma patient-derived xenograft animal model (PDX) was established in female BALB/c nude mice. The tumor tissue of osteosarcoma was obtained, cut into small fragment, and transplanted subcutaneously into BALB/c nude mice.

When the tumor volumes reached approximately 100 mm³, twenty female BALB/c nude mice bearing 143B tumors were randomly assigned into four groups. The mice were injected intravenously with PBS, Ru(II)-OH, NP2, NP4 (2 mg/kg Ru) on days 1, 4, 7, 10, respectively. The tumor volume and body weight of all the mice were recorded every two days. The tumor volume (in mm³) was calculated using the formula V = (a × b²)/2, where "a" represents the length of the tumor and "b" denotes its width.

### Histopathological analysis

The solid tumor and major organs were harvested from tumor-bearing mice for histological observation by standard H&E and immunofluorescence staining. For H&E staining, the excised tumor and organs were fixed in 4% paraformaldehyde solution, embedded in paraffin, sectioned, and stained with H&E. The nuclei were counterstained using DAPI (Thermo Fisher Scientific), after which the stained sections were imaged with a confocal microscope (LSM880, Zeiss).

### Statistical analysis

All statistical analyses were conducted by utilizing GraphPad Prism 10 (GraphPad Software). Data are expressed as mean ± standard deviation (SD) from at least three independent experiments unless otherwise specified in the figure legend. Statistical significance between pairs of groups was assessed using an unpaired two-sided t-test, one-way ANOVA followed by the Dunnett multiple comparison test, or two-way ANOVA with the Tukey multiple comparison test, as indicated. The number of replicates performed is detailed in each figure legend, where applicable.

### Reporting summary

Further information on research design is available in the Nature Portfolio Reporting Summary linked to this article.

## Data availability

The nuclear magnetic resonance and mass spectrometry data are provided in the Supplementary Information. All other data can be found within the main text, Supplementary information or Source Data files. Source data is provided with this paper. Source data are provided with this paper.

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

## Acknowledgements

J. Karges gratefully acknowledges the financial support provided by the Liebig fellowship from the Chemical Industry Fund of the German Chemical Industry Association, the Life Sciences Bridge Award from the Aventis Foundation, and the Paul Ehrlich & Ludwig Darmstaedter Early Career Award 2024 – a prize awarded by the Paul Ehrlich Foundation, Germany. This work is supported by the National Key Research & Development Program (2022YFC2603900), Beijing Natural Science Foundation (JQ24055), and the National Natural Science Foundation of China (22175189, 52373127). The schematic illustration in Figs. 7a, 8a, 9a, 10a is created with BioRender.com.

## Author contributions

All authors were involved with the design and interpretation of experiments and with the writing of the manuscript. Chemical, physical, and biological experiments were carried out by H. Zhang, N. Montesdeoca, D. Tang, G. Liang, M. Cui, C. Xu, L.-M. Servos, T. Bing, Z. Papadopoulos, and M. Shen. The work was supervised by H. Xiao, Y. Yu and J. Karges. All authors have given approval to the final version of the manuscript.

## Funding

## Competing interests

The authors declare no competing interests.
