## [Peer Review File · Nature Communications]

REVIEWER COMMENTS

Reviewer #1 (ICD, apoptosis):

Significance

The manuscript by Hanchen Zhang and co-workers describes a novel ruthenium-based cancer drug that is also effective under hypoxic conditions. Many solid tumours have a hypoxic core when they reach a size of more than 400 μ m in diameter. These cores are usually characterized by remnants of secondary necrotic tumour cells that have died due to a lack of oxygen and nutrients. These already dead tumour cells do not need any treatment. However, tumour cells at the edge of the necrotic core suffer from sublethal starvation and therefore activate a variety of different stress response mechanisms, which as a side effect also make them resistant to various drugs.

The distinction between the necrotic core and its sub-lethal hypoxic rim containing stressed tumour cells should be described in the introduction section.

In addition to the induction of resistance mechanisms as response to sublethal hypoxia, also delivery, uptake and chemical conversion can be reduced under these conditions. There is therefore an urgent need to overcome these barriers. The current manuscript focuses on the oxygen-dependency of metal based cancer drugs. The authors developed ruthenium-containing nanoreactors for the catalytic oxidation of GSH to GSSG in hypoxic tumours. As GSH is essential to keep the cellular redox potential, the uptake of this drug will finally lead to cell death. Thus, the manuscript deals with a clinically relevant and timely question.

Note that hypoxia-inducible factors switch tumour cells from oxidative to glycolytic metabolism, to reduce mitochondrial superoxide generation, and increase the synthesis of NADPH and GSH, in order to maintain redox homeostasis under hypoxic conditions.

The authors should discuss the extent to which such mechanisms may antagonise the effect of the ruthenium drug

Study design

The authors hypothesised that the proposed ruthenium catalyst will maintain its anti-cancer activity even under hypoxic conditions. Therefore, they first calculated the potential ability of different ruthenium catalysts to convert GSH into GSSG. Then they confirmed the results of a selected ruthenium catalyst with in vitro experiments. Then the ruthenium catalyst was linked to a polymer to mediate the specific delivery to tumour cells. Its anti-cancer activity was verified in

osteosarcoma cell lines as well as with an in vivo model of osteosarcoma. This study design is suitable for investigating the hypothesis.

Methodology

Since I am no expert in this field I do not evaluate the methodology of the chapters “Design, Molecular Synthesis, and Characterization”, “Catalytic Activity”, and “Encapsulation into Nanoparticles”.

The authors cultivated osteosarcoma cells at 21% and 1% O₂ and termed these conditions “normoxic” and “hypoxic”, respectively. However, the usual P-O₂ in the tissue is less than 40mm Hg as compared to 160mm Hg in the air, i.e. compared to the 21% O₂ in the air we have around 5% in the tissue. In the brain, it can be even lower. Thus, 21% is hyperoxic and do not correspond to the situation of a normoxic tissue. The difference between normoxic (5% or 40mm Hg) and hypoxic (1% or 8mm Hg) is much smaller than the text suggests. This has to be indicated in the result section and discussed. The term “normoxic” for 21% O₂ should be removed.

Figure 5a shows the uptake of the Cy5.5 labelled drug at different z-axis distances. In the last image, we see a lower staining of the core. What is the reason? Are the cells in the core still viable? What is the estimated pO₂ within this core?

The higher cytotoxic effect of NP4 under hypoxic conditions shown in Figure 5D-G are convincing.

Cell death is always associated with a complete destruction of the anti-oxidative defense of a cell. Thus, it has to be expected that a higher cell death in presence of NP4 under hypoxic conditions shown in Figure 5, results in a harsh decrease of the GSH/GSSG ratio (as shown in Figure 6A) and consequently to an increase of ROS (as shown in Figure 6B-C). On the other hand, a collapse of the GSH/GSSG ratio would definitively lead to cell death. From the association between cell death with the GSH/GSSG ratio we cannot conclude what is the hen and what the egg. The main hint that NP4 induces under hypoxic conditions first a decrease of GSH/GSSG and then cell death comes from the GSH-degrading catalytic activity shown in Figure 3. The authors should investigate whether the NP4-induced cell death/or the decrease in GSH/GSSG ratio can be prevented by anti-oxidative compounds. E.g., they could repeat the experiments shown in Figure 5 and 6A-C with different concentrations of NP4 in the presence or absence of such compounds. This would increase the informative value of the experiments on causality.

Macrophages react sensitively to damage associated molecular patterns (DAMPs) released from dead cells, especially from necrotic cells. According to the annexin/PI assay shown in Figure 5F,

NP4 induces the highest number of necrotic cells (i.e. FITC-A negative / PC5.5A positive) of all treatments. Thus, it is not surprising that macrophages exposed to NP4-treated cells secrete the highest amount of TNFalpha (Fig. 7F). Here again, it should be investigated whether this could be prevented by an anti-oxidative treatment.

Conclusions

The anti-tumour effect of NP4 is impressive and convincing. However, the evidence provided that this is mediated by the consumption of GSH under hypoxic conditions is not convincing. Further experiments are required to proof this.

Reviewer #2 (Osteosarcoma model/therapy):

The present study would be much more valuable and relevant if patient derived orthotopic mouse models of osteosarcoma were used (PMID:34848441).

Reviewer #3 (Catalytic Metallodrugs, nanomedicine):

In this article, the authors incorporated ruthenium catalyst into tumor-targeting polymeric nanoreactors for hypoxic anticancer therapy. The nanoparticles could not only disrupt intracellular redox homeostasis but also stimulate macrophage activation, leading to a great impairment of tumor growth. The authors designed the nanoparticles from eight concerns. However, they did not provide powerful results to demonstrate their conclusions. Major concerns are raised as follows.

--More experimental groups should be set for completeness. In Figure S21, why did the authors choose 10 mM H₂O₂ to detect ROS-triggered drug release? In Figure 5a, the cellular uptake of NP4 is not good after 7h incubation, the authors should extend the incubation time. Moreover, the cellular uptake of nanoparticles should be detected in both K7M2 cells and 143B cells under hypoxia and normoxia conditions. The macrophage activation should also be detected in BMDM. Besides, in some figures like Figures 8 b to d, the authors merely set one experimental group. It cannot verify the superiority of nanoparticles. Though the metal-based catalytic drugs do not show fluorescent signals, they can be measured through ICP-MS.

--More basic data should be provided in the therapeutic efficacy section. For example, time-dependent body-weight curves, relative BL intensity, survival rate, and H&E staining of tumor tissues after treatment should be provided in the orthotopic K7M2 osteosarcoma model for completeness. Images of tumors and tumor weight should be provided in the 143B tumor-bearing mouse model. Moreover, the biological mechanism of NP4 mentioned previously should also be investigated in vivo. The authors demonstrated that NP4 could trigger a cascade of immune response, why did they verify the therapeutic efficacy in immunodeficient mice? The pharmacokinetics profiles of nanoparticles should be provided to verify the long circulation time.

--More additional details should be provided in the section of "Methods". For example, it is not clear how to prepare the Cy5-labeled nanoparticles, how to build a hypoxic environment, how to load oxygen into nanoparticles, and so on. Moreover, some basic details like the number of cells cultured and the concentration of the treated drug should also be clearly provided. The author should thoroughly record their methods to support their results in the manuscript.

--The release mechanism of Ru(III)-OH from oligomeric units should be elaborated. The intracellular drug release should be monitored.

--The oxygen loading capacity of NP4 should be detected. The stability of NP4 in vitro should be investigated.

-- Why is the cytotoxicity of NP4 in a hypoxic environment stronger than that under a normoxic environment?

--The writing requires substantial revision in order to ensure smoother reading. It is hard to understand like "...lead to cell death inside cancer cells".

We thank the Reviewers for their very positive feedbacks and their suggestions/comments regarding the submission of our article. Below, we provide a point-by-point reply to each of their comments/suggestions. For your information, the changes made to the document are highlighted in yellow.

Reviewer 1:

Significance

The manuscript by Hanchen Zhang and co-workers describes a novel ruthenium-based cancer drug that is also effective under hypoxic conditions. Many solid tumours have a hypoxic core when they reach a size of more than 400 μ m in diameter. These cores are usually characterized by remnants of secondary necrotic tumour cells that have died due to a lack of oxygen and nutrients. These already dead tumour cells do not need any treatment. However, tumour cells at the edge of the necrotic core suffer from sublethal starvation and therefore activate a variety of different stress response mechanisms, which as a side effect also make them resistant to various drugs.

Response: We thank the reviewer for the positive feedback and highlighting the significance of this study.

The distinction between the necrotic core and its sub-lethal hypoxic rim containing stressed tumour cells should be described in the introduction section.

Response: We thank the reviewers for bringing this to our attention. We agree with the reviewer. Solid tumors have an inadequate supply of oxygen within them due to the rapid proliferation of tumor cells, incomplete vascularization, and uneven distribution, and ultimately result in the presence of oxygen-deprived areas within the tumor. In the core, a necrotic nucleus is formed due to the long-term lack of oxygen and nutrients. The necrotic nucleus extends outward into a hypoxic region characterized by low oxygen concentrations. We have added this information into the revised version of the manuscript. It reads "Solid tumors suffer from an inadequate oxygen supply due to the rapid proliferation of cancer cells, incomplete vascularization, and uneven distribution. This results in oxygen-deprived areas within the tumor tissue. In the core, a necrotic nucleus forms due to the prolonged lack of oxygen and

nutrients. The necrotic nucleus extends outward into a region with low oxygen concentrations, referred to as hypoxia. “

In addition to the induction of resistance mechanisms as response to sublethal hypoxia, also delivery, uptake and chemical conversion can be reduced under these conditions. There is therefore an urgent need to overcome these barriers. The current manuscript focuses on the oxygen-dependency of metal based cancer drugs. The authors developed ruthenium-containing nanoreactors for the catalytic oxidation of GSH to GSSG in hypoxic tumours. As GSH is essential to keep the cellular redox potential, the uptake of this drug will finally lead to cell death. Thus, the manuscript deals with a clinically relevant and timely question.

Response: We thank the reviewer for highlighting the significance and timeliness of this study.

Note that hypoxia-inducible factors switch tumour cells from oxidative to glycolytic metabolism, to reduce mitochondrial superoxide generation, and increase the synthesis of NADPH and GSH, in order to maintain redox homeostasis under hypoxic conditions. The authors should discuss the extent to which such mechanisms may antagonise the effect of the ruthenium drug.

Response: We thank the reviewer for pointing this out. We agree with the reviewer that cancer cells are changing their metabolism within a hypoxic environment. However, these are exactly the conditions our herein reported therapeutic formulation excels in. The herein reported metal complex oxidizes GSH into GSSG inside cancerous cells and therefore can interact in cancer cells with different oxygen concentrations. Importantly, the compound does not interact in a stoichiometric way but instead functions as a catalyst. As such it can convert high concentrations of GSH inside the cancer cell and function even under hypoxic conditions.

Study design

The authors hypothesised that the proposed ruthenium catalyst will maintain its anti-cancer activity even under hypoxic conditions. Therefore, they first calculated the potential ability of different ruthenium catalysts to convert GSH into GSSG. Then they confirmed the results of a selected ruthenium catalyst with in vitro experiments. Then the ruthenium catalyst was linked to a polymer to mediate the specific delivery to tumour cells. Its anti-cancer activity was verified in osteosarcoma cell lines as well as with an in vivo model of osteosarcoma. This study design is suitable for investigating the hypothesis.

Response: Thanks to the reviewer for recognizing the elaborate design as well as highlighting the multidisciplinary approach taken to thoroughly evaluate the new compounds.

Methodology

Since I am no expert in this field I do not evaluate the methodology of the chapters “Design, Molecular Synthesis, and Characterization”, “Catalytic Activity”, and “Encapsulation into Nanoparticles”. The authors cultivated osteosarcoma cells at 21% and 1% O₂ and termed these conditions “normoxic” and “hypoxic”, respectively. However, the usual P-O₂ in the tissue is less than 40mm Hg as compared to 160mm Hg in the air, i.e. compared to the 21% O₂ in the air we have around 5% in the tissue. In the brain, it can be even lower. Thus, 21% is hyperoxic and do not correspond to the situation of a normoxic tissue. The difference between normoxic (5% or 40mm Hg) and hypoxic (1% or 8mm Hg) is much smaller than the text suggests. This has to be indicated in the result section and discussed. The term “normoxic” for 21% O₂ should be removed.

Response: We thank the reviewers for bringing this to our attention. We agree with the reviewer and have changed these claims within the revised manuscript. We have removed the term “normoxic” from the revised manuscript. In addition, we have added a statement explaining that the studied oxygen levels do not represent the concentrations inside a human patient but are merely a model to study the therapeutic effect at high and low oxygen concentrations. We have added this information into the revised version of the manuscript. It reads: “The biological properties of NP4 were in-depth studied against human osteosarcoma (143B) and mouse osteosarcoma (K7M2) cells with different O₂ concentrations. As a model for physiological conditions the cancer cells were cultivated and treated with a 21% O₂ atmosphere, and as a model for hypoxic conditions the cancer cells were cultivated and treated with a 1% O₂ atmosphere. It is important to mention that these O₂ levels do not represent the concentrations found inside human healthy and cancerous tissues and are followingly used to investigate the different therapeutic efficiency of the nanoparticle formulation in high and low O₂ atmospheres.”

Figure 5a shows the uptake of the Cy5.5 labelled drug at different z-axis distances. In the last image, we see a lower staining of the core. What is the reason?

Response: We thank the reviewer for pointing this out. We have studied the cellular uptake of the nanoparticles upon labelling it with a Cy5.5 fluorophore and monitoring of the emission inside the multicellular tumor spheroid. Based on the increasing depth of the spheroid and the detection limit of our microscopy system, we are unable to evaluate the cellular penetration of deeper levels of the multicellular tumor spheroids.

Are the cells in the core still viable? What is the estimated pO₂ within this core?

Response: We thank the reviewer for bringing this to our attention. The whole spheroid is viable and consist of living cells. At the presented size, the multicellular spheroid does not have a hypoxic core. For the establishment of a hypoxic center, the multicellular spheroid would need to be grown to a bigger size as previously described in the literature.

The higher cytotoxic effect of NP4 under hypoxic conditions shown in Figure 5D-G are convincing.

Response: We thank the reviewer for highlighting this key finding of this study.

Cell death is always associated with a complete destruction of the anti-oxidative defense of a cell. Thus, it has to be expected that a higher cell death in presence of NP4 under hypoxic conditions shown in Figure 5, results in a harsh decrease of the GSH/GSSG ratio (as shown in Figure 6A) and consequently to an increase of ROS (as shown in Figure 6B-C). On the other hand, a collapse of the GSH/GSSG ratio would definitively lead to cell death. From the association between cell death with the GSH/GSSG ratio we cannot conclude what is the hen and what the egg. The main hint that NP4 induces under hypoxic conditions first a decrease of GSH/GSSG and then cell death comes from the GSH-degrading catalytic activity shown in Figure 3. The authors should investigate whether the NP4-induced cell death/or the decrease in GSH/GSSG ratio can be prevented by anti-oxidative compounds. E.g., they could repeat the experiments shown in Figure 5 and 6A-C with different concentrations of NP4 in the presence or absence of such compounds. This would increase the informative value of the experiments on causality.

Response: We thank the reviewer for pointing this out. We agree with the reviewer and have therefore studied the cytotoxicity of the nanoparticle in the presence of the anti-oxidant N-acetylcysteine (NAC). The results showed that the incubation with NAC was able to prevent the oxidation of GSH to GSSG

(Figure R1). The reduced conversion of GSH to GSSG resulted in the decreased generation of ROS, which were monitored by flow cytometry (Figure R2) and fluorescence microscopy (Figure R3). Capitalizing on this, also a reduced cytotoxicity upon incubation with the anti-oxidant NAC was observed (Figure R4). We have added this information into the revised version of the manuscript. It reads “Interestingly, the incubation with the antioxidant N-acetylcysteine upon treatment with NP4 resulted in reduced levels of apoptosis, suggestive that the cytotoxicity of NP4 is related to intracellular redox homeostasis (Fig S26) [...] The ability to catalytically convert GSH into GSSG inside the cancer cells was assessed upon determination of the levels of GSH and GSSG after treatment. The cancer cells that were incubated with PBS were found with a GSH/GSSG ratio of 1.55. In contrast, upon treatment with NP4 in a 1% O₂ atmosphere, the GSH/GSSG ratio dropped to 0.12. Upon preincubation of the cancer cells with the antioxidant N-acetylcysteine and subsequent treatment with NP4 the GSH/GSSG ratio was significantly less influenced, reaching a GSH/GSSG ratio of 0.71 (Fig. 5a), suggestive of the importance of the redox hemostasis for the depletion of GSH from the cancer cells [...] For a deeper understanding, the cancer cells were preincubated with NAC and the presence of ROS upon treatment assessed using the ROS specific probe 2',7'-dichlorodihydrofluorescein diacetate. While the treatment with NP4 alone caused a strong production of ROS, the preincubation with NAC and subsequent treatment with NP4 showed a poor ROS generation (Fig. 5d) [...] Following the semi qualitative assessment, the ROS levels were quantified by flow cytometry. The results demonstrated a very strong accumulation of ROS inside the hypoxic cancer cells upon treatment with NP4 but a poor ROS accumulation upon preincubation with NAC and subsequent treatment with NP4 (flow cytometry plots: Fig. 5b, quantification: Fig. 5c).”

Figure R1. Cellular GSH/GSSG ratio inside the cancer cells upon various treatments.

Figure R2. Flow cytometry plots and quantification of 143B cells under hypoxic conditions treated with of Ru(II)-OH, NP2, NP4 and NP4+NAC (10 μ M Ru, 20 mM NAC) for 24 h and stained with 2',7'-dichlorodihydrofluorescein diacetate.

Figure R3. Representative CLSM images of 143B cells under hypoxic conditions treated with of Ru(II)-OH, NP2, NP4 and NP4+NAC (10 μ M Ru, 20 mM NAC) for 24 h and stained with the RPS probe 2',7'-dichlorodihydrofluorescein diacetate (DCFH-DA, green) and DAPI (blue, nucleus).

Figure R4. Flow cytometry plots of 143B cells under hypoxic conditions treated with of Ru(II)-OH, NP2, NP4 and NP4 +NAC (10 μ M Ru, 20 mM NAC) for 24 h and stained with Annexin V-FITC and propidium iodide.

Macrophages react sensitively to damage associated molecular patterns (DAMPs) released from dead cells, especially from necrotic cells. According to the annexin/PI assay shown in Figure 5F, NP4 induces the highest number of necrotic cells (i.e. FITC-A negative / PC5.5A positive) of all treatments. Thus, it is not surprising that macrophages exposed to NP4-treated cells secrete the highest amount of TNFalpha (Fig. 7F). Here again, it should be investigated whether this could be prevented by an anti-oxidative treatment.

Response: We thank the reviewer for bringing this to our attention. We agree with the reviewer and we have investigated whether macrophage secretion of TNF- α could be blocked by the antioxidant NAC. The experimental results showed a significant decrease in TNF- α secretion by macrophages pretreated in advance with NAC, suggesting that macrophage activation is associated with NP2-catalyzed production of reactive oxygen species (ROS). This figure has been added as Figure 7h in the revised manuscript, and the relevant statements have been added the revised version of the manuscript. It reads: “The cancer cells preincubated with NAC and subsequently treated with NP4 showed a reduced number of TNF- α secreted from the macrophages, suggesting that macrophage activation is associated with the NP4-induced oxidative stress.”

Figure R5. The concentration of TNF- α in RAW264.7 macrophage supernatants was determined by ELISA upon various treatments.

Conclusions

The anti-tumour effect of NP4 is impressive and convincing. However, the evidence provided that this is mediated by the consumption of GSH under hypoxic conditions is not convincing. Further experiments are required to proof this.

Response: We thank the reviewer for bringing this to our attention. In agreement with the previous comment of this reviewer we have therefore studied the cytotoxicity of the nanoparticle in the presence of the anti-oxidant N-acetylcysteine (NAC). The results showed that the incubation with NAC was able to prevent the oxidation of GSH to GSSG (Figure R1). The reduced conversion of GSH to GSSG resulted in the decreased generation of ROS, which were monitored by flow cytometry (Figure R2) and fluorescence microscopy (Figure R3). Capitalizing on this, also a reduced cytotoxicity upon incubation with the anti-oxidant NAC was observed (Figure R4). The assessment of the GSH levels inside the treated animal model is challenging. As the depletion of GSH results in the rise of the ROS levels, we have assessed the ROS levels in the tumor tissue of NP4-treated mice. This figure has been added as Figure S38 in the revised supporting information. The relevant statements have been added the revised version of the manuscript. It reads “For an additional insight into the mechanism of action, the tumorous tissue was stained with a ROS-specific dye and analyzed by CLSM. While the animal model treated with PBS did not show any signs for ROS, high amounts of fluorescence signals in the tissue from the animal treated with NP4 were observed (Fig S38).”

Figure R6. ROS dyes (red) and DAPI (blue) stain of tumor tissue slices after various treatments

Reviewer 2:

The present study would be much more valuable and relevant if patient derived orthotopic mouse models of osteosarcoma were used (PMID:34848441).

Response: We thank the reviewer for pointing this out. As recommended by the reviewer, we have established a patient-derived osteosarcoma xenograft model and studied the therapeutic efficiency of NP4 in this animal model. These findings have been added to the revised manuscript. It reads: “For a better understanding of the clinical potential, its therapeutic properties were assessed in a patient-derived osteosarcoma xenograft model (PDXOS) (protocol for model establishment: Fig 10a). While the tumors treated with PBS showed an exponential growth, the tumor of the animals treated with Ru(II)-OH, NP2, or NP4 showed a tumor growth inhibition effect. Importantly, the treatment with NP4 showed the strongest therapeutic effect (individual tumor growth inhibition curves: Fig 10b, combined tumor growth inhibition curves: Fig 10c). Importantly, the animal models treated with the nanoparticles did not show any signs for pain, stress, or discomfort, and did not lose any weight (Fig 10d), suggestive of the high biocompatibility of the treatment. 15 days after the treatment, the animal models were sacrificed and the tumorous tissue obtained (photographs of the tumor: Fig 10e, tumor weight: Fig 10f), confirming the strong therapeutic effect of NP4. Histological hematoxylin and eosin staining of the tumorous tissue of the animal model treated with NP4 showed high amounts of cell death and nuclear fragmentation (Fig 10g, top). Terminal deoxynucleotidyl transferase dUTP nick end labelling staining further demonstrated high number of apoptotic cells inside the tumorous tissue of the animal model treated with NP4 (Fig 10g, bottom). For an additional insight into the mechanism of action, the tumorous tissue was stained with a ROS-specific dye and analysed by CLSM. While the animal model treated with PBS did not show any signs for ROS, high amounts of fluorescence signals in the tissue from the animal treated with NP4 were observed (Fig S38). Combined these findings demonstrated the high potential of the NP4 to eradicate challenging patient-derived osteosarcoma tumors inside an animal model.”

Figure R7. Evaluation of the therapeutic properties of Ru(II)-OH, NP2, or NP4 (2 mg Ru/kg) upon intravenous injection in the osteosarcoma PDX model. a) Schematic treatment schedule. b) c) Tumor growth inhibition curves upon treatment. d) Body weight upon treatment. e) Representative tumor images and f) Tumor weight of different treatment groups after 14 days. $n = 5$ mice per group. g) H&E and TUNEL staining of tumor tissues. Data are presented as mean \pm standard deviation. Statistical significance between every two groups was calculated by *t*-test.

Reviewer 3:

In this article, the authors incorporated ruthenium catalyst into tumor-targeting polymeric nanoreactors for hypoxic anticancer therapy. The nanoparticles could not only disrupt intracellular redox homeostasis but also stimulate macrophage activation, leading to a great impairment of tumor growth. The authors designed the nanoparticles from eight concerns. However, they did not provide powerful results to demonstrate their conclusions. Major concerns are raised as follows.

Response: We thank the reviewers for their valuable comments and will respond to each of the following comments

More experimental groups should be set for completeness.

Response: We thank the reviewer for bringing this to our attention. We are unsure which experiment the reviewer is referring to. Within the revised manuscript we have ensured that each experiment has been repeated at least three-times (typically even more repeats have been performed). All experiments were also statistically analyzed to ensure the soundness of the claims in this manuscript

In Figure S21, why did the authors choose 10 mM H₂O₂ to detect ROS-triggered drug release?

Response: We thank the reviewer for pointing this out. We have studied the stability of the nanoparticles upon incubation in PBS and PBS supplemented with hydrogen peroxide. Hydrogen peroxide is commonly used in the chemical literature as a model for ROS (e.g. Nat Commun 14, 5350 (2023). <https://doi.org/10.1038/s41467-023-40826-5>). This information has been added to the revised manuscript. It reads: “. To investigate this potential release, NP4 was incubated with phosphate-buffered saline (PBS) or PBS containing H₂O₂ as a model for ROS inside the cancer cells and the release monitored by ICP-MS. While NP4 remained relatively stable under physiological conditions, the payload was readily released in the presence of H₂O₂ (Fig S22).”

In Figure 5a, the cellular uptake of NP4 is not good after 7h incubation, the authors should extend the incubation time.

Response: We thank the reviewer for bringing this to our attention. As an important property, the cellular uptake of the nanoparticles in I43B cells was investigated upon labelling of NP4 with the well-characterized dye Cy5.5 into the nanoparticle formulation (denoted as Cy5.5-NP4). The time-dependent monitoring by confocal laser scanning microscopy (CLSM) indicated that with a prolongation of the incubation time (from 1 h to 12 h), an increasing amount of Cy5.5-NP4 was detected inside the cancer cells (Fig. R8a). Complementary, this finding was confirmed by flow cytometry (flow cytometry plots: Fig. R8b, quantification: Fig. R8c). The updated CLSM and flow cytometry results was provided in the revised manuscript. This information has been added to the revised manuscript. It reads: “The cellular uptake of the nanoparticles in I43B cells was investigated upon labelling of NP4 with the well-characterized dye Cy5.5 into the nanoparticle formulation NP-Cy5.5. The time-dependent monitoring by confocal laser scanning microscopy (CLSM) indicated that with a prolongation of the incubation time, an increasing amount of NP-Cy5.5 was detected inside the cancer cells (Fig 4a). Complementary, this finding was confirmed by flow cytometry (flow cytometry plots: Fig 4b, quantification: Fig 4c).”

Figure R8. a) Time-dependent CLSM images of the cellular uptake of Cy5.5-NP4 in 143B cells or 143B multicellular tumor spheroids determined by CLSM. The cell nucleus was stained with 4', 6-diamidino-2-phenylindol (DAPI) and the cell cytoskeleton was stained with Alexa-488. b) Time-dependent cellular uptake of Cy5.5-NP4 in 143B cells by determined flow cytometry. c) Quantification of the cellular uptake of Cy5.5-NP4 from b.

Moreover, the cellular uptake of nanoparticles should be detected in both K7M2 cells and 143B cells under hypoxia and normoxia conditions.

Response: We thank the reviewer for pointing this out. The cellular uptake of nanoparticles was detected in both K7M2 cells and 143B cells under hypoxia and normoxia conditions by flow cytometry. The results showed that no significant dependence of cellular uptake of nanoparticles on oxygen content was observed in the cells. This figure has been added as Figure S24 in the revised supporting information and Figure 5b-c in the revised manuscript, and the relevant statements have been added in the revised manuscript. It reads “Cellular uptake studies were performed in cellular environments with different oxygen concentrations in the cell lines 143B and K7M2. The results showed no significant dependence of the cellular uptake of nanoparticles on the oxygen concentration in the cells (Fig S24).”

Figure R9. The cellular uptake of nanoparticles was detected in both K7M2 cells and 143B cells under hypoxia and normoxia conditions by flow cytometry.

The macrophage activation should also be detected in BMDM.

Response: We thank the reviewer for bringing this to our attention. The macrophage activation was detected in BMDM. The experimental results showed that NP4-treated tumor cells were able to promote the differentiation of tumor-associated macrophages to pro-inflammatory M1-type macrophages. This figure has been added as Figure S29 in the revised supporting information, and the relevant statements have been added in the revised manuscript. It reads “To validate macrophage activation, mouse bone marrow-derived macrophages were extracted and incubated with cancer cells, that have been treated with the nanoparticles. The macrophage activation was analyzed by flow cytometry. The results showed that the treated cancer cells promoted the differentiation of the mouse bone marrow-derived macrophages. The treatment with NP4 enhanced the level of pro-inflammatory M1 macrophages by 1.97 in comparison to the treatment with NP2. In addition, the treatment with NP4 reduced the level of anti-inflammatory M2-type macrophages by 0.20 in comparison to the treatment with NP2 (Fig S29).”

Figure R10. The differentiation of tumor-associated macrophages.

Besides, in some figures like Figures 8 b to d, the authors merely set one experimental group. It cannot verify the superiority of nanoparticles. Though the metal-based catalytic drugs do not show fluorescent signals, they can be measured through ICP-MS.

Response: We thank the reviewer for pointing this out. Within the revised manuscript we have ensured that each experiment has been repeated at least three-times (typically even more repeats have been performed). All experiments were also statistically analyzed to ensure the soundness of the claims in this manuscript. If possible, we have studied the cellular properties using different types of analytic techniques including ICP-MS, as suggested by the reviewer. Exemplary, we have studied the biodistribution inside the animal model upon intravenous injection in the tail vein. The experimental results showed that after the compound was injected, the nanoparticles had a high tumor accumulation than the molecular agent. The figure has been added as Figure S32 in the revised supporting information. This information has been added to the revised manuscript. It reads: “Complementary, the high tumor accumulation was confirmed upon determination of the metal content in the respective organs by ICP-MS. The direct comparison upon using the same concentration of ruthenium of the molecular agent Ru(II)-OH and NP4 showed a significantly higher tumor accumulation of the nanoparticle formulation NP4 (Fig 32).”

Figure R11. Ruthenium levels in different organs and tumor sites 48 h after intravenous injection of NP4 and Ru(II)-OH.

More basic data should be provided in the therapeutic efficacy section. For example, time-dependent body-weight curves, relative BL intensity, survival rate, and H&E staining of tumor tissues after treatment should be provided in the orthotopic K7M2 osteosarcoma model for completeness.

Response: We thank the reviewer for pointing this out. We agree with the reviewer and we have provided all suggested data in the revised manuscript or supporting information. For example, the figure of time-dependent body-weight curves was added as the Figure S34 in the revised supporting information. The figure of relative BL intensity was added as the Figure S33 in the revised supporting information. The figure of H&E staining of tumor tissues after treatment was added as the Figure S35 in the revised supporting information. Since the treatment of the mice was studied for only 12 days, all animals survived and there is no particular record on the survival rate of the mice for this study. This information has been added in the revised manuscript. It reads “Thus, the therapeutic effect of NP4 on the orthotopic K7M2-Luc osteosarcoma mouse model was evaluated (Fig. 8e, quantification: Fig. 33) [...] During the treatment, no significant change in the body weight of the mice was observed, indicating that NP4 has a high biocompatibility (Fig S34) [...] This conclusion was further confirmed by H&E and TUNEL staining (Fig. 8f, Fig. 35), which revealed that the tumor tissues of mice treated with NP4 exhibited the strongest green fluorescence (apoptotic cells), demonstrating the outstanding anti-tumor activity of NP4.”

Figure R12. Body weight upon treatment in orthotopic K7M2 osteosarcoma model.

Figure R13. Relative BL intensity upon treatment in orthotopic K7M2 osteosarcoma model.

Figure R14. H&E staining of tumor tissues after treatment in orthotopic K7M2 osteosarcoma model.

Images of tumors and tumor weight should be provided in the 143B tumor-bearing mouse model.

Response: We thank the reviewer for bringing this to our attention. We agree with the reviewer and we have added the images of the tumors and the tumor weight in the revised manuscript or supporting information. This information has been added in the revised manuscript. It reads “After the treatment, the mice were sacrificed, the tumor tissues were obtained, and analyzed (photographs of the tumor: Fig S37, tumor weight: Fig 9g). The tumor of the animals treated with NP4 showed a strong tumor growth inhibition effect.”

Figure R15. Photograph of tumors isolated from mice.”

Figure R16. Average tumor weight after the mice were sacrificed on day 14.

Moreover, the biological mechanism of NP4 mentioned previously should also be investigated in vivo. *Response:* We thank the reviewer for pointing this out. As recommended by the reviewer, we have established a patient-derived osteosarcoma xenograft model and studied the therapeutic efficiency of NP4 in this animal model. These findings have been added to the revised manuscript. It reads: “For a better understanding of the clinical potential, its therapeutic properties were assessed in a patient-derived osteosarcoma xenograft model (PDXOS) (protocol for model establishment: Fig 10a). While the tumors treated with PBS showed an exponential growth, the tumor of the animals treated with Ru(II)-OH, NP2, or NP4 showed a tumor growth inhibition effect. Importantly, the treatment with NP4 showed the strongest therapeutic effect (individual tumor growth inhibition curves: Fig 10b, combined tumor growth inhibition curves: Fig 10c). Importantly, the animal models treated with the nanoparticles did not show any signs for pain, stress, or discomfort, and did not lose any weight (Fig 10d), suggestive of the high biocompatibility of the treatment. 15 days after the treatment, the animal models were sacrificed and the tumorous tissue obtained (photographs of the tumor: Fig 10e, tumor weight: Fig 10f), confirming the strong therapeutic effect of NP4. Histological hematoxylin and eosin staining of the tumorous tissue of the animal model treated with NP4 showed high amounts of cell death and nuclear fragmentation (Fig 10g, top). Terminal deoxynucleotidyl transferase dUTP nick end labelling staining further demonstrated high number of apoptotic cells inside the tumorous tissue of the animal model treated with NP4 (Fig 10g, bottom). For an additional insight into the mechanism of action, the tumorous tissue was stained with a ROS-specific dye and analysed by CLSM. While the animal model treated with PBS did not show any signs for ROS, high amounts of fluorescence signals in the tissue from the animal treated with NP4 were observed (Fig S38). Combined these findings

demonstrated the high potential of the NP4 to eradicate challenging patient-derived osteosarcoma tumors inside an animal model.”

Figure R6. ROS dyes (red) and DAPI (blue) stain of tumor tissue slices after various treatments

The authors demonstrated that NP4 could trigger a cascade of immune response, why did they verify the therapeutic efficacy in immunodeficient mice?

Response: We thank the reviewer for pointing this out. We agree with the reviewer that the presence or deficiency of the immune system of the animal model has a tremendous effect on the ability to activate the immune response. We have studied the therapeutic efficiency of the nanoparticles in a murine K7M2 osteosarcoma animal model with an intact immune system. Importantly, the whole immunotherapeutic evaluation was done in a murine K7M2 osteosarcoma animal model with an intact immune system. Following this evaluation, the therapeutic efficiency was studied in a human 143B osteosarcoma animal model and patient-derived osteosarcoma xenograft mouse model with an immunodeficient immune system. This was necessary to prevent the rejection of the human tumor in the mouse animal model. As such, we have studied the therapeutic efficiency and immunotherapeutic effect with murine osteosarcoma cells in a mouse model with an intact immune system and the therapeutic efficiency with human osteosarcoma cells in a mouse model with an immunodeficient immune system. We have added this information to the revised manuscript.

The pharmacokinetics profiles of nanoparticles should be provided to verify the long circulation time.

Response: We thank the reviewer for bringing this to our attention. We agree with the reviewer and we have studied the pharmacokinetic profile of the nanoparticles. The nanoparticles were found with a enhanced blood circulation time than the molecular agent. This figure has been added as Figure S31 in the revised supporting information. This information has been added in the revision manuscript. It reads “The blood circulation of the nanoparticles was assed upon intravenous injection into the tail vein of healthy female mice and determination of the metal content in the blood by ICP-MS. As expected, the molecular metal complex Ru(II)-OH was quickly cleared from the blood stream with a blood circulation half-life time of 0.28 h. In contrast, the nanoparticles NP4 had a much longer blood circulation half-life time of 2.97 h (Fig S31), suggesting a higher bioavailability of NP4 in comparison to Ru(II)-OH.”

Figure R17. Pharmacokinetic profile of Ru(II)-OH and NP4.

More additional details should be provided in the section of “Methods”. For example, it is not clear how to prepare the Cy5-labeled nanoparticles,

Response: We thank the reviewer for pointing this out. The Cy5-labeled nanoparticles prepared by encapsulating the hydrophobic fluorescent dye Cy5.5-NHS via the amphiphilic polymer P4 using hydrophilic-hydrophobic interaction. More additional details had been provided in the section of “Methods”.

how to build a hypoxic environment,

Response: We thank the reviewer for bringing this to our attention. The hypoxic environment is constructed in a hypoxic incubator, and the oxygen concentration in the incubator is adjusted to meet the experimental requirements.

how to load oxygen into nanoparticles, and so on.

Response: We thank the reviewer for pointing this out. Since the intermolecular forces of perfluorinated compounds are very low, the gaps between the molecules are large and oxygen is easily carried. As such the polymer P4 has the potential to transport oxygen due to large amounts of perfluorinated moieties in the polymer backbone. Nitrogen gas was bubbled into aqueous solutions of PBS, NP2 and NP4 to exclude any air in solution. Afterwards, oxygen gas was bubbled into the aqueous solutions for 5 minutes to achieve oxygen storage. This experimental protocol has been previously described in various publications such as *Adv. Mater.* 2024, 36, 2308780. <https://doi.org/10.1002/adma.202308780>; *J. Am. Chem. Soc.* 2024, 146, 2, 1644-1656. <https://doi.org/10.1021/jacs.3c12416>; *J. Am. Chem. Soc.* 2024, 146, 11, 7543-7554. <https://doi.org/10.1021/jacs.3c13501>.

Moreover, some basic details like the number of cells cultured and the concentration of the treated drug should also be clearly provided.

Response: We thank the reviewer for bringing this to our attention. We have thoroughly revised the methods section and added these information.

The author should thoroughly record their methods to support their results in the manuscript.

Response: We thank the reviewer for bringing this to our attention. We have thoroughly revised the methods section and added these information.

For example:

Preparation of fluorescently labeled nanoparticles: Dyes (Cy5.5, Cy7.5 or Nile Red) and P4 (100 mg) were co-dissolved in 1 mL DMSO, and the solution were then slowly added dropwise to 10 mL of water. Unloaded dyes and organic solvent were removed through dialysis against deionized water for 48 h. The obtained NP-Cy5.5, NP-Cy7.5, and NP-NR were stored at 4°C for subsequent use.

Measurement of oxygen loading capacity of NP4: PBS, NP2 and NP4 were firstly bubbled with the flowing N₂ to exclude the content of O₂. Afterward, the PBS, NP2 and NP4 solutions were bubbled with the flowing O₂, where the O₂ concentration was monitored using a portable dissolved oxygen meter (Hanna S20 Instruments HI-2004-02 Edge® Dissolved Oxygen Meter) for 5 min. The values were recorded automatically every 30 s.

Pharmacokinetic assay of NP4: The pharmacokinetics on Ru(II)-OH and NP4 (Ru 800 μM, 200 μL) were evaluated in healthy BALB/c mice. Six mice were randomly divided into two groups and injected into the tail vein with Ru(II)-OH and NP4, respectively, and 10 μL of blood was collected from the tail tip at 5 min, 10 min, 20 min, 30 min, 1 h, 2 h, 4 h, 8 h, 12 h, 24 h, and 48 h. Finally, the blood samples were nitrated with nitric acid, and the ruthenium content was determined by ICP-MS.

Tumor model establishment and biodistribution study: To establish orthotopic model, K7M2 cells (3×10⁶ cells/well) were dispersed in PBS buffer and implanted into right tibia of BALB/c mice. One week after the tumor was implanted, the tumor volume reached to about 100 mm³. the mice were i.v. injected with PBS, Ru(II)-OH, NP2, and NP4 at the dose of 2 mg Ru/kg. The mice were sacrificed after 48 h and the heart, liver, spleen, lungs, kidneys, and tumor tissues of the mice were isolated, weighed and nitrated. Finally, the ruthenium content in each organ and tumor tissue was determined by ICP-MS.

The release mechanism of Ru(III)-OH from oligomeric units should be elaborated. The intracellular drug release should be monitored.

Response: We thank the reviewer for pointing this out. The intracellular release mechanism of Ru(III)-OH from the nanoparticles is experimentally difficult to be monitored directly. Therefore, we encapsulated the fluorescent dye Nile Red into the nanoparticles (NP4-NR). When Nile Red is in the hydrophobic core of the nanoparticles, it shows a strong fluorescent signal. In contrast upon release into the hydrophilic cytoplasm, the fluorescence is quenched. We observed the fluorescence changes of NP4 encapsulating Nile red dye inside the cell by fluorescence microscopy. The experimental results showed that the intracellular fluorescence peaked when NP4-NR was incubated with tumor cells for 4 h. And in the following 10 h, the intracellular red fluorescence gradually weakened until it disappeared. It indicated that the nanoparticles were gradually degraded inside the cells and Nile red was gradually released from the nanoparticles. Similarly, the Ru(III)-OH can be released from the nanoparticles. This figure has been added as Figure S25 in the revised supporting information. The relevant statements have been added in the revision manuscript. It reads "The biodegradability of NP4 upon internalization into the cancer cells was studied upon encapsulation of the dye Nile Red into a nanoparticle formulation, followingly referred to as NP4-NR. When Nile Red is in the hydrophobic core of the nanoparticles, it is highly fluorescent and can be easily detected. However, upon release in the hydrophilic cytoplasm, the fluorescence of the dye is quenched, allowing for the facile detection of the degradation of the nanoparticles. CLSM images upon incubation of NP4-NR with the cancer cells showed the steadily increase of the fluorescence inside the cancer cells, reaching a maximal emission after incubation for 4 h. Further monitoring of the cancer cells for an additional 10 h

demonstrated the gradual decrease of the red fluorescence signal, indicative of the degradation of the nanoparticles inside the cancer cells (Fig S25).”

Figure R18. CLSM images of 143B cells incubated with NP4-NR for different time.

The oxygen loading capacity of NP4 should be detected.

Response: We thank the reviewer for bringing this to our attention. The oxygen loading capacity of NP4 was studied using a dissolved oxygen analyzer. The oxygen concentrations have been determined as previously reported in the literature such as *Adv. Mater.* 2024, 36, 2308780. <https://doi.org/10.1002/adma.202308780>; *J. Am. Chem. Soc.* 2024, 146, 2, 1644-1656. <https://doi.org/10.1021/jacs.3c12416>; *J. Am. Chem. Soc.* 2024, 146, 11, 7543-7554. <https://doi.org/10.1021/jacs.3c13501>. The results showed that NP4 with a perfluoro moiety has a higher oxygen loading capacity than NP2. This figure has been added as Figure S23 in the revised supporting information. The information has been added to the revision manuscript. It reads: “The ability of the nanoparticles to store and transport oxygen was studied using a dissolved oxygen analyzer. Oxygen gas was bubbled through aqueous solutions of NP2 and NP4 and the amount of trapped oxygen time-dependently quantified. NP4 reached a maximal oxygen storage of 53 mg/L, which was approximately 2.5 more than for NP2 (Fig. S23), indicative of the oxygen store and transport properties of the carbon fluorinated nanoparticles NP4.”

Figure R19. Determination of oxygen content in solution.

Adv. Mater. 2024, 36, 2308780.

J. Am. Chem. Soc. 2024, 146, 1644-1656.

J. Am. Chem. Soc. 2024, 146, 7543-7554.

Figure R20. Determination of oxygen content in references.

The stability of NP4 in vitro should be investigated.

Response: We thank the reviewer for bringing this to our attention. The stability of nanoparticles is an important characteristic of nanoparticles and an important prerequisite for their application to living systems. Thus, the stability of nanoparticles NP4 over 14 days was evaluated by dynamic light scattering measurements. The results showed that the particle size and potential of NP4 remained relatively stable over 14 days. This figure has been added as Figure S21 in the revised supporting information. This information has been added to the revision manuscript. It reads: “Meanwhile, the stability of nanoparticles was evaluated using DLS. The results showed that the average hydrodynamic diameter and PDI of NP4 remained unchanged over a 14-day period, indicative of the excellent stability of NP4 (Fig. S21).”

Figure R21. The stability assessment of NP4 upon incubation of PBS via DLS.

Why is the cytotoxicity of NP4 in a hypoxic environment stronger than that under a normoxic environment?

Response: We thank the reviewer for pointing out this aspect. Cancer cells that are cultivated in normoxic or hypoxic environments have significantly different metabolisms. Exemplary, we note that hypoxia-inducible factors switch tumor cells from oxidative to glycolytic metabolism, to reduce mitochondrial superoxide generation, and increase the synthesis of NADPH and GSH, in order to maintain redox homeostasis under hypoxic conditions. These fundamentally different conditions result in different treatment efficiencies. Interestingly, the hypoxic conditions of the cancer cells are the

conditions our herein reported therapeutic formulation excels in. The herein reported metal complex oxidizes GSH into GSSG inside cancerous cells and therefore can interact in cancer cells with different oxygen concentrations. Importantly, the compound does not interact in a stoichiometric way but instead functions as a catalyst. As such it can convert high concentrations of GSH inside the cancer cell and function even under hypoxic conditions.

The writing requires substantial revision in order to ensure smoother reading. It is hard to understand like "...lead to cell death inside cancer cells"

Response: We thank the reviewer for bringing this to our attention. We have carefully revised and corrected the manuscript to avoid grammatical and writing errors. For example, revise sentence "Among the most promising therapeutic approaches, much research interest has been devoted to catalytic metal-based anticancer agents that could convert natural biomolecules into cytotoxic products or products that ultimately lead to cell death inside cancer cells" to "Among the most promising therapeutic approaches, much research interest has been devoted to catalytic metal-based anticancer agents that could convert natural biomolecules into cytotoxic products that ultimately lead to cell death."

We do think that the modifications and clarifications present in the revised version of the manuscript have significantly improved the manuscript and, importantly, that we have responded adequately to all major concerns of the Reviewers. We strongly believe that the results of this study will be of high interest to the readers of *Nature Communications*.

With my best regards,

REVIEWERS' COMMENTS

Reviewer #1:

The authors answered all my questions satisfactorily.

Reviewer #3:

The manuscript has been revised mostly according to the comments and suggestions of the reviewers, I recommend the acceptance of the manuscript for publication in Nature Communicatoins.